# FedTT: Cross-City Federated Traffic Knowledge Transfer with Privacy Preservation

## Abstract

Traffic prediction (TP) is a core task in urban computing, aiming to forecast future traffic conditions from historical observations. To overcome the scarcity of traffic data in emerging cities, recent studies have explored Federated Traffic Knowledge Transfer (FTT), which leverages data-rich source cities to assist data-scarce target cities without raw data sharing. However, existing FTT approaches are limited by three unresolved challenges: (i) potential *privacy leakage* since gradients or parameters generated during federated computing can still be inverted, (ii) severe *cross-city distribution discrepancies* that reduce transfer effectiveness, and (iii) *low data quality* caused by missing or unreliable sensor readings. To address these challenges, we propose **FedTT**, a novel federated framework for cross-city traffic knowledge transfer with privacy-preserving. FedTT introduces three innovations: (i) a lightweight **Traffic Secret Aggregation (TSA)** protocol that achieves secure knowledge aggregation without sacrificing efficiency or accuracy; (ii) a **Traffic Domain Adapter (TDA)** that explicitly aligns heterogeneous source–target distributions for more effective transfer, and (iii) a **Traffic View Imputation (TVI)** method that leverages spatio-temporal dependencies to complete missing traffic data robustly. Extensive experiments on four real-world datasets show that FedTT achieves significant improvements over 18 state-of-the-art baselines, consistently reducing prediction error while maintaining strong privacy protection.

## 1 Introduction

**Traffic Prediction (TP)** (Qin et al., 2024; Zhao et al., 2023) aims to forecast traffic conditions based on historical traffic data (e.g., flow, speed, and occupancy). It is a fundamental task in urban computing, supporting congestion management (Yuan et al., 2022) and the allocation of public resources (Meng et al., 2021). Although many powerful TP models have been developed (Ji et al., 2023; Jiang et al., 2023), they typically require large volumes of high-quality traffic data for model training. In practice, however, many cities—especially those with newly deployed or incomplete sensing infrastructures—suffer from data scarcity (Wang et al., 2019; 2022a), which makes it difficult to train reliable models and often leads to overfitting (Jin et al., 2023; Mo & Gong, 2023).

To alleviate data scarcity, **Transfer Learning (TL)** has been widely adopted to transfer knowledge from data-rich source cities to data-scarce target cities (Liu et al., 2023; Lu et al., 2022; Tang et al., 2022). However, most existing TL methods rely on centralized frameworks that require the exchange of raw traffic data, which poses significant privacy risks (Liu et al., 2020; Meng et al., 2021; Yang et al., 2024). Although traffic flow data may appear to contain only aggregated statistics (e.g., volume counts per region), such information can still be exploited by malicious actors to infer sensitive details—such as individual vehicle trajectories or location patterns—through data linkage and reconstruction attacks (Akin et al., 2025; Chen et al., 2024a). Moreover, stringent data protection regulations like the GDPR (2016) and CCPA (2018) explicitly restrict the transfer of personally identifiable information. These legal constraints make direct inter-city data transmission infeasible in practice. As illustrated in Fig. 1(a), with datasets such

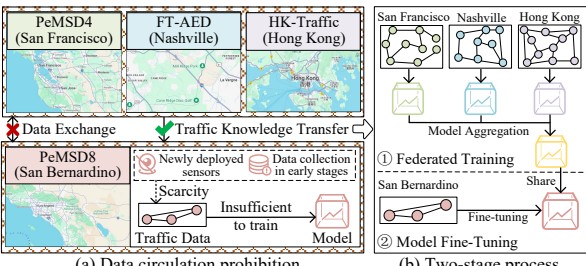

Figure 1: Privacy-preserving traffic knowledge transfer

as PeMSD4 (PeMS, 2024) (San Francisco, SF), FT-AED (Coursey et al., 2024) (Nashville, NV), HK-Traffic (Hong Kong, HK (2024)), and PeMSD8 (PeMS, 2024) (San Bernardino, SB), SF, NV, and HK serve as source cities, while SB is the target. Under current regulatory frameworks, these cities cannot share their local traffic data directly, limiting each to its own isolated data repository.

**Federated Learning (FL)** (Wang et al., 2024c; Liu et al., 2024a; Yang et al., 2024) has emerged as a promising paradigm to preserve privacy by training models collaboratively without raw data sharing, and has already been deployed in urban computing applications (Wang et al., 2022b; Gu et al., 2020). Motivated by this, recent studies (Qi et al., 2023; Zhang et al., 2024b) have explored FL for **Federated Traffic Knowledge Transfer (FTT)**. As illustrated in Fig. 1(b), mainstream FTT methods adopt a two-stage pipeline: (i) source cities (e.g., SF, NV, and HK) jointly train a global model via FL, and (ii) the global model is fine-tuned on the target city (e.g., SB). While effective in certain cases, this paradigm leaves three **unresolved fundamental challenges**.

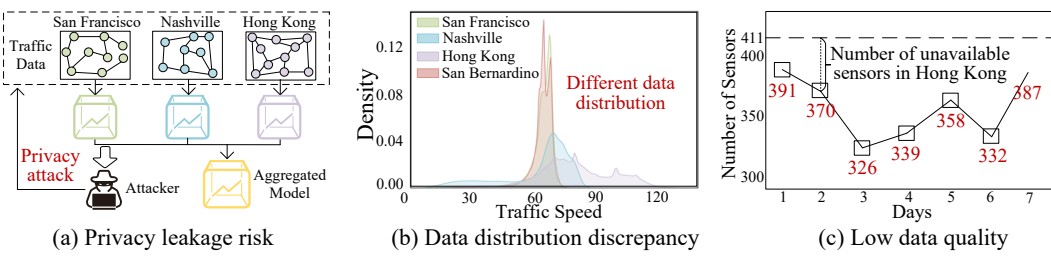

(a) Privacy leakage risk  (b) Data distribution discrepancy  (c) Low data quality

Figure 2: Three unresolved challenges in federated traffic knowledge transfer (FTT)

**C1: How to effectively protect data privacy in FTT?** Although federated learning (FL) avoids direct raw data exchange, existing FTT approaches still suffer from potential privacy leakage. This is because existing FTT methods require uploading gradients or model parameters to the server, which are vulnerable to inference attacks that can recover sensitive information (Gao et al., 2024; Wang et al., 2024b; Zheng et al., 2024), as shown in Fig. 2(a). A natural remedy is to adopt privacy-preserving techniques such as **Homomorphic Encryption (HE)** (Rivest et al., 1978) or **Differential Privacy (DP)** (Dwork et al., 2006). However, HE introduces significant computation and communication overheads, while DP degrades data utility and model accuracy (Wang et al., 2024a; Tawose et al., 2023). Existing works stop short of balancing privacy, efficiency, and accuracy in FTT. Therefore, how to design a lightweight yet effective privacy-preserving mechanism tailored to federated traffic knowledge transfer remains an open problem.

**C2: How to mitigate the impact of cross-city data distribution discrepancies on FTT?** Traffic data distributions vary drastically across cities due to differences in road networks, traffic patterns, and sensor deployments. Although there are transfer learning methods designed to minimize domain shifts in other applications, they are not directly applicable here. Most existing approaches consider that feature spaces are only weakly shifted, but in traffic data, the heterogeneity is much stronger: two cities may have entirely different distributions. For example, Fig. 2(b) shows that NV and SB exhibit distinct domains. More importantly, prior FTT studies (Liu et al., 2023; Lu et al., 2022; Tang et al., 2022) largely overlook such cross-city discrepancies, thereby undermining transfer effectiveness. Thus, a key challenge is how to adaptively align heterogeneous traffic domains in a way that respects spatio-temporal structures, which goes beyond generic domain adaptation techniques.

**C3: How to overcome low traffic data quality issues in FTT?** Real-world traffic data are often incomplete due to sensor failures or communication losses (Yuan et al., 2024; Qin et al., 2021). Specifically, traffic data present domain-specific challenges: missing values are not random but often correlated with spatio-temporal factors such as peak hours, road congestion, or sensor location. As illustrated in Fig. 2(c), the number of available sensors in HK fluctuates considerably, introducing instability into model training. Conventional imputation methods (Peng et al., 2023; Yuan et al., 2024) fill in missing values but fail to fully capture the complex spatio-temporal dependencies, resulting in suboptimal predictions. Therefore, robust traffic knowledge transfer requires imputation strategies that are explicitly spatio-temporal aware, which has not been addressed in prior FTT literature. Addressing this gap is crucial to ensuring the robustness and practicality of FTT models.

**Contributions.** To the best of our knowledge, no prior studies address the above challenges in a unified federated setting. To this end, we propose **FedTT**, a novel Federated learning framework for cross-city Traffic knowledge Transfer. Specifically, FedTT introduces three key and novel modules, which are explicitly designed to address these challenges, respectively.

- **(C1 → TSA)** We design a **Traffic Secret Aggregation (TSA)** protocol that enables secure aggregation of transferred knowledge without exposing raw data, gradients, or parameters. Unlike prior works that rely on heavy cryptographic primitives (e.g., homomorphic encryption or differential privacy), TSA achieves a better trade-off between privacy, efficiency, and accuracy.

- **(C2 → TDA)** We introduce a **Traffic Domain Adapter (TDA)** to explicitly address cross-city domain discrepancies. By transforming, aligning, and classifying traffic data from source to target domains, TDA ensures that transferred knowledge is more consistent and generalizable across heterogeneous urban environments.

- **(C3 → TVI)** We propose a **Traffic View Imputation (TVI)** method to handle incomplete traffic data. TVI leverages spatio-temporal dependencies for missing data completion, thereby enhancing the robustness of federated transfer in real-world scenarios with noisy or unreliable sensors.

- As a system-level optimization, we develop a **Federated Parallel Training (FPT)** module (**Appendix A.1**) to improve training efficiency. FPT reduces communication overhead and increases parallelism through split learning and parallel optimization, enabling scalable and practical knowledge transfer in real-world federated deployments.

- **Extensive experiments** on four real-world datasets demonstrate that FedTT consistently outperforms 18 state-of-the-art baselines in terms of prediction accuracy, privacy preservation, communication efficiency, runtime, and scalability.

## 2 RELATED WORK

### 2.1 TRAFFIC PREDICTION

Traffic prediction plays a critical role in the development of smart cities and has garnered significant attention in the spatio-temporal data mining community. For instance, ST-SSL (Ji et al., 2023) improves traffic pattern representation to account for spatial and temporal heterogeneity through a self-supervised learning framework. DyHSL (Zhao et al., 2023) leverages hypergraph structure information to model the dynamics of a traffic network, updating the representation of each node by aggregating messages from associated hyperedges. Additionally, PDFormer (Jiang et al., 2023) introduces a spatial self-attention module to capture dynamic spatial dependencies and a flow-delay-aware feature transformation module to model the time delays in spatial information propagation.

Since these models are centralized and require uploading raw traffic data to a server, several studies (Yuan et al., 2022; Lai et al., 2023; Xia et al., 2023; Li & Liu, 2024; Yang et al., 2024; Liu et al., 2024b) explore federated learning for privacy-preserving traffic prediction. Representative examples include FedGRU (Liu et al., 2020), which integrates GRU with federated averaging, and CNFGNN (Meng et al., 2021), which separates temporal and spatial modeling across devices and servers. However, when traffic data in emerging cities is scarce or incomplete, federated models may overfit local data and fail to provide accurate predictions. ***In contrast, we focus on enabling data-scarce cities to benefit from traffic knowledge in data-rich cities through a federated and privacy-preserving transfer framework.***

### 2.2 TRAFFIC KNOWLEDGE TRANSFER

Traffic knowledge transfer methods can be grouped into (1) single-source transfer, (2) multi-source transfer, and (3) federated traffic transfer.

Single-source transfer (STT). Early work (Jin et al., 2023; Ouyang et al., 2024; Mo & Gong, 2023) learns transferable representations from one source city to a target city. Although effective under moderate distribution gaps, these methods degrade significantly when the source–target differences are large. Multi-source transfer (MTT). Subsequent studies (Yao et al., 2019; Liu et al., 2021; Zhang et al., 2024b) leverage multiple source cities to provide more diverse knowledge. For example, TPB (Liu et al., 2023) builds a traffic pattern bank, and DastNet (Tang et al., 2022) obtains domain-invariant embeddings via domain adaptation. However, these approaches rely on centralized data sharing, which raises privacy concerns. Federated traffic transfer (FTT). Recent works, including T-ISTGNN (Qi et al., 2023), pFedCTP (Zhang et al., 2024b), and 2MGTCN (Yuan et al., 2025), aim to conduct cross-city transfer under federated settings. Despite important progress, they still suffer from challenges such as privacy leakage, distribution discrepancies, low-quality data, and high transfer overhead, limiting their practicality in real-world deployments. ***In contrast, we aim***

*to propose a privacy-preserving and efficient federated learning framework for cross-city traffic knowledge transfer to address the challenges of privacy, effectiveness, and robustness in FTT.*

## 3 PROBLEM DEFINITIONS

Table 1: Notations and descriptions

| Notation | Description |
|---|---|
| $m, \mathcal{M}$ | A sensor and a set of sensors $\{m_1, m_2, \ldots\}$ |
| $\mathcal{E}, A$ | A set of edges and the weighted adjacent matrix of edges |
| $\mathcal{G}$ | A road network $(\mathcal{M}, \mathcal{E}, A)$ |
| $t, r, tr$ | The time, $r$-th, and training round |
| $M_t$ | A set of available sensors $\{m_i \mid i \leq |\mathcal{M}|\}$ at time $t$ |
| $X_t, X_{(r)}$ | The traffic data at time $t$ and the $r$-th traffic data |
| $F_1$ | The dimension of the traffic data features |
| $\mathcal{X}, D$ | A set of traffic data $\{X_1, X_2, \ldots\}$ and a traffic dataset $\{X_1, X_2, \ldots; \mathcal{G}\}$ |
| $c, s$ | A client and the server |
| $R, S$ | A source city and the target city |
| $n$ | The number of clients and source cities |
| $\mathcal{C}, \mathcal{R}$ | A set of clients $\{c_1, c_2, \ldots, c_n\}$ and source cities $\{R_1, R_2, \ldots, R_n\}$ |
| $\theta, \mathcal{L}(\cdot)$ | A model and a loss function |
| $v_t^i, V_t$ | The $i$-level traffic subview and a traffic view $\{v_t^1, v_t^2, \ldots\}$ at time $t$ |
| $\mathcal{P}$ | A traffic domain prototype |

The frequently used notations and descriptions in this paper are shown in Table 1.

**Definition 1 (Road Network).** *The road network is a weighted graph $\mathcal{G} = (\mathcal{M}, \mathcal{E}, A)$, where $\mathcal{M} = \{m_1, m_2, \ldots\}$ is the set of sensors, $\mathcal{E} \subseteq \mathcal{M} \times \mathcal{M}$ is the set of edges, and $A \in \mathbb{R}^{|\mathcal{M}| \times |\mathcal{M}|}$ is the weighted adjacency matrix of edges. Here, $m_i$ denotes the sensor with index $i$.*

**Definition 2 (Traffic Data).** *Given the available sensors $M_t = \{m_i \mid i \leq |\mathcal{M}|\}$, the traffic data is denoted as $\mathcal{X} = \{X_1, X_2, \ldots\}$, where $X_t \in \mathbb{R}^{|M_t| \times F_1}$ is the traffic data of $|M_t|$ available sensors at time $t$. Here, $F_1$ denotes the number of traffic data features. For instance, $F_1 = 3$ when the traffic data includes flow, speed, and occupancy data, which are the number of vehicles, the average speed of vehicles, and the percentage of time a sensor detects vehicles over a period of time, respectively.*

**Problem Formulation (FTT).** In federated learning, multiple clients $\mathcal{C} = \{c_1, c_2, \ldots, c_n\}$ collaboratively train a global model using their local data. In the first stage, FTT trains a traffic model $\theta_{TP}$ to learn traffic knowledge from source cities $\mathcal{R} = \{R_1, R_2, \ldots, R_n\}$, where each source city $R_i$ corresponds to a client $c_i$, as formally shown below:

$$\min_{\theta_{TP}} \frac{1}{n} \sum_{i=1}^{n} \mathcal{L}(\theta_{TP}, D^{R_i}), \tag{1}$$

where $\mathcal{L}(\cdot)$ is the loss function, and $D^{R_i} = \{X_1^{R_i}, X_2^{R_i}, \ldots; \mathcal{G}^{R_i}\}$ is the traffic dataset of the source city $R_i$. Here, $\mathcal{G}^{R_i}$ and $X_t^{R_i}$ are the road network and the traffic data at time $t$ of the source city $R_i$. In the second stage, given target city' dataset $D^S = \{X_1^S, X_2^S, \ldots; \mathcal{G}^S\}$, FTT predicts the next $T'$ traffic data based on the $T$ historical observations at time $t$ in the target city $S$, as shown below:

$$\{X_{t-T+1}^S, X_{t-T+2}^S, ..., X_t^S; \mathcal{G}^S\} \xrightarrow{\theta_{TP}} \{X_{t+1}^S, X_{t+2}^S, ..., X_{t+T'}^S\} \tag{2}$$

## 4 OUR METHODS

Fig. 3 illustrates the architecture of the proposed FedTT framework, which comprises three modules: Traffic View Imputation (TVI), Traffic Domain Adapter (TDA), and Traffic Secret Aggregation (TSA). As shown in Fig. 3(a), FedTT comprises $n$ clients $\mathcal{C} = \{c_1, c_2, \ldots, c_n\}$ and a central server $s$.

Specifically, each source city $R_i$ is treated as a client $c_i$, while the target city $S$ is treated as the server $s$. The traffic domains of the data in clients are transformed to align with the server's domain, and the server's traffic model is trained on this transformed data uploaded by clients. Consequently, the FTT problem defined in Eqs. 1 and 2 is reformulated to minimize the sum of the following losses:

$$\min_{\theta_{TP}} \frac{1}{n} \sum_{i=1}^{n} \mathcal{L}(\theta_{TP}, D^{R_i \to S}, D^S), \tag{3}$$

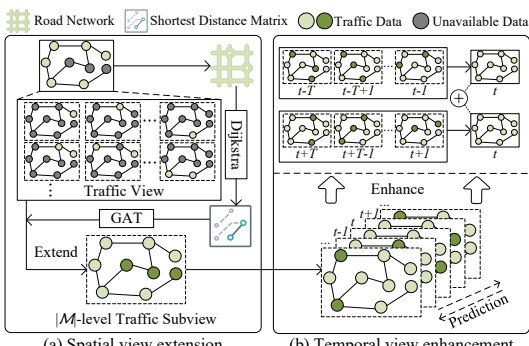

Figure 3: The architecture of the proposed FedTT framework

where $D^{R_i \to S}$ represents the traffic dataset whose domain is transformed from the source city $R_i$ to the target city $S$. The overall process of FedTDP is as following. First, the TVI module captures spatial and temporal dependencies within the traffic data to extend and enhance the traffic view (①–②), as shown in Fig. 3(b). Then, the TDA module conducts traffic domain transformation and alignment for the source cities' data (③–④). Besides, the module performs traffic domain classification to categorize the traffic data domain (⑤), as shown in Fig. 3(c). Finally, the TSA module employs the proposed traffic secret aggregation method to securely mask and aggregate the transformed data from source cities (⑥–⑦), as shown in Fig. 3(d). The target of our FedTT is to transfer traffic knowledge across cities while preserving privacy, handling data discrepancies and low data quality challenges.

## 4.1 TRAFFIC VIEW IMPUTATION

**Design Motivation.** Existing federated traffic transfer methods often overlook the challenges associated with low-quality traffic data, especially when missing data is prevalent, thereby significantly undermining the performance of traffic knowledge transfer models. Although some data augmentation methods (Chen et al., 2024b; Peng et al., 2023; Yuan et al., 2024) can be leveraged for imputation, they fail to effectively capture the spatio-temporal dependencies of data, leading to suboptimal accuracy. In contrast, we propose the Traffic View Imputation (TVI) method to enhance traffic data quality by completing missing traffic data through a comprehensive exploration of the spatial and temporal dependencies inherent in traffic data:

Figure 4: The process of traffic view imputation

$$\{X_1, X_2, \ldots; \mathcal{G}\} \xrightarrow{\theta_{TVI}} \{\widetilde{X}_1, \widetilde{X}_2, \ldots\}, \tag{4}$$

where $\theta_{TVI}$ is the TVI model consisting of a spatial view extension model $\theta_{SV}$ and a temporal view enhancement model $\theta_{TV}$. Besides, $\widetilde{X}_t$ is the imputed traffic data of all sensors. In addition, the traffic view represents the traffic data of all sensors at a certain time, as defined below.

**Definition 3 (Traffic View).** *A traffic view is the snapshot of traffic data of sensors $\mathcal{M}$ at time $t$, consisting of a set of multi-level traffic subviews, denoted as $V_t = \{v_t^1, v_t^2, \ldots v_t^{|M_t|}\}$, where $i$-level traffic subview $v_t^i$ is a set of traffic data of $i$ sensors at time $t$.*

**i) Spatial View Extension.** In the first stage, TVI extends the $|\mathcal{M}|$-level traffic subview at time $t$:

$$\{v_t^1, v_t^2, \ldots v_t^{|M_t|}; \mathcal{G}\} \xrightarrow{\theta_{SV}} sv_t^{|\mathcal{M}|}, \tag{5}$$

where $\theta_{SV}$ denotes the spatial view extension model and $sv_t^{|\mathcal{M}|}$ represents the extended $|\mathcal{M}|$-level traffic subview at time $t$. As shown in Fig. 4(a), it first computes the shortest distance matrix $\mathcal{A} = \{A_1, A_2, \ldots, A_{|\mathcal{M}|}\}$, where $A_i$ represents the shortest distance tensor of sensor $m_i$ to other sensors. This is computed using Dijkstra (2022) algorithm with the weighted adjacency matrix $A$. Next, the feature of each sensor is computed, i.e., $h_i = \theta_{GAT}(A_i)$, where $h_i$ represents the $K$-head feature of sensor $m_i$ with $F_2$ feature dimensions, and $\theta_{GAT}$ is the Graph Attention Network (GAT) model with $K = 8$ and $F_2 = 128$. Additionally, the extension of multi-level traffic subviews is

averaged to obtain the $|\mathcal{M}|$-level traffic subview with a Multi-Layer Perception (MLP) $\theta_E$:

$$sv_t^{|\mathcal{M}|} = \frac{1}{|V_t|} \sum_{i=1}^{|V_t|} \frac{1}{|v_t^i|} \sum_{j=1}^{|v_t^i|} \theta_E\left(\frac{1}{i} \sum_{k=1}^{i} (H(v_t^i[j][k]) \cdot (v_t^i[j][k])^\top)\right), \quad (6)$$

where $v_t^i[j][k]$ represents the traffic data of the $k$-th sensor in the $j$-th combination within the $i$-level traffic subview at time $t$, and $H(v_t^i[j][k]) \in \mathbb{R}^{K \times F_2 \times 1}$ represents the multi-head feature of the sensor corresponding to $v_t^i[j][k]$. Finally, it computes the loss of available sensors to train the $\theta_{SV}$ model:

$$\min_{\theta_{SV}} \mathcal{L}(\theta_{SV}, \mathcal{V}_{SV}) = \min_{\theta_{SV}} \frac{1}{|\mathcal{V}_{SV}|} \sum_{t=1}^{|\mathcal{V}_{SV}|} \frac{1}{|M_t|} (sv_t^{|M_t|} - X_t), \quad (7)$$

where $\mathcal{V}_{SV} = \{sv_1^{|\mathcal{M}|}, sv_2^{|\mathcal{M}|}, \ldots\}$ is the set of extended traffic subviews at different times, and $sv_t^{|M_t|}$ is the predicted traffic data of available sensors at time $t$.

**ii) Temporal View Enhancement.** As shown in Fig. 4(b), in the second stage, TVI enhances the $|\mathcal{M}|$-level traffic subview based on the preceding/succeeding $|\mathcal{M}|$-level traffic subviews:

$$\begin{aligned} \{sv_{t-T}^{|\mathcal{M}|}, sv_{t-T+1}^{|\mathcal{M}|}, \ldots, sv_{t-1}^{|\mathcal{M}|}\} &\xrightarrow{\theta_{TV}} tv_t^{|\mathcal{M}|}, \\ \{sv_{t+T}^{|\mathcal{M}|}, sv_{t+T-1}^{|\mathcal{M}|}, \ldots, sv_{t+1}^{|\mathcal{M}|}\} &\xrightarrow{\theta_{TV}} tv_t^{|\mathcal{M}|}, \end{aligned} \quad (8)$$

where $tv_t^{|\mathcal{M}|}$ represents the enhanced $|\mathcal{M}|$-level traffic subview, whose final value is the average of the above two results. Besides, $\theta_{TV}$ is the temporal view enhancement model that employs the SOTA DyHSL traffic model (Zhao et al., 2023). Then, it computes the loss of available sensors to train $\theta_{TV}$:

$$\min_{\theta_{TV}} \mathcal{L}(\theta_{TV}, V^{|\mathcal{M}|}) = \min_{\theta_{TV}} \frac{1}{|V^{|\mathcal{M}|}|} \sum_{t=1}^{|V^{|\mathcal{M}|}|} \frac{1}{|M_t|} (tv_t^{|M_t|} - X_t), \quad (9)$$

where $\mathcal{V}_{TV} = \{tv_1^{|\mathcal{M}|}, tv_2^{|\mathcal{M}|}, \ldots\}$ represents the set of enhanced traffic subviews and $tv_t^{|M_t|}$ is the predicted traffic data of the available sensors at time $t$. Finally, we get the predicted traffic data of all $|\mathcal{M}|$ sensors $\widetilde{X}_t = tv_t^{|\mathcal{M}|}$. Note that the training of the TVI model is completed before the training of the FedTT framework, as it only needs to be conducted within each city.

## 4.2 Traffic Domain Adapter

**Design Motivation.** None of the existing approaches consider traffic data distribution discrepancies between the source and target cities in FTT, which decreases the effectiveness of traffic knowledge transfer. Motivated by this, to reduce the impact of traffic data distribution discrepancies on model performance, we propose the Traffic Domain Adapter (TDA) module, as shown in Fig. 5. This module reduces traffic domain discrepancies by uniformly transforming data from the traffic domain of the source city ("source domain" for short) to the traffic domain of the target city ("target domain" for short):

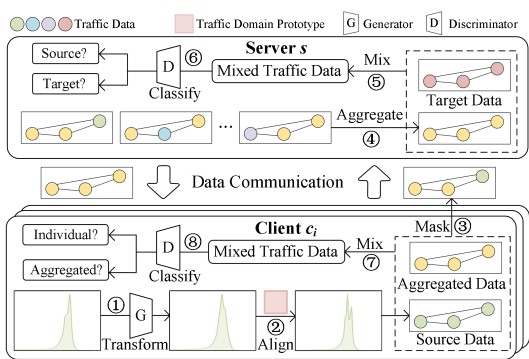

Figure 5: TDA and TSA modules

$$\{\widetilde{X}_1^R, \widetilde{X}_2^R, \ldots\} \xrightarrow{\theta_{TDA}} \{X_1^{R \to S}, X_2^{R \to S}, \ldots\}, \quad (10)$$

where $X_t^{R \to S}$ is the transformed data of $|\mathcal{M}^S|$ sensors, and $\theta_{TDA}$ is a generative adversarial network (Wang et al., 2018) consisting of a generator model $\theta_{Gen}$ and a discriminator model $\theta_{Dis}$.

**i) Traffic Domain Transformation.** In the first step, TDA uses the generator model, road network, and traffic domain prototype to transform the traffic data from the source domain to the target domain, as shown in Fig. 5 (①), where the traffic domain prototype is the representative traffic sample that can reflect the main feature of traffic data in the domain, as formally defined below.

**Definition 4 (Traffic Domain Prototype).** *Given the traffic data $\mathcal{X} = \{X_1, X_2, \ldots\}$ in a traffic domain, a traffic domain prototype $\mathcal{P}$ is the central traffic data, which is computed as the averaged value of all traffic data, i.e., $\mathcal{P} = \frac{1}{|\mathcal{X}|} \sum_{t=1}^{|\mathcal{X}|} X_t$.*

First, it computes the transformation matrix $A_{\mathcal{G}}$ of the road network through $\left(A_{\mathcal{G}}\right)^{\top} \cdot \mathcal{G}^R \cdot A_{\mathcal{G}} = \mathcal{G}^S$, where $A_{\mathcal{G}}$ can learn the road network information of the source and target cities, which is computed by the gradient descent method. Similarly, it then computes the transformation matrix $A_{\mathcal{P}}$ of the traffic domain prototype through $A_{\mathcal{P}} \cdot \mathcal{P}^R = \mathcal{P}^S$, where $\mathcal{P}^R$ and $\mathcal{P}^S$ are traffic domain prototypes of the source and target cities, respectively. Here, $A_{\mathcal{P}}$ can learn the traffic domain prototype information of the source and target cities, which is computed by the gradient descent method. Then, the generator model leverages $A_{\mathcal{G}}$ and $A_{\mathcal{P}}$ to transform the traffic data using MLP models $\theta_{\mathcal{G}}$, $\theta_{\mathcal{P}}$, and $\theta_X$:

$$X_t^{R \to S} = \theta_{\mathcal{G}}(A_{\mathcal{G}} \cdot \widetilde{X}_t^R) + \theta_{\mathcal{P}}(A_{\mathcal{P}} \cdot \widetilde{X}_t^R) + \theta_X(\widetilde{X}_t^R), \tag{11}$$

**ii) Traffic Domain Alignment.** In the second step, TDA trains the generator model $\theta_{Gen}$, as shown in Fig. 5 (②). Specifically, it aligns the transformed data $\mathcal{X}^{R \to S} = \{\widetilde{X}_1^{R \to S}, X_2^{R \to S}, \ldots\}$ of the source city with the traffic domain prototype $\mathcal{P}^S$ of the target city $S$, as described below:

$$\min_{\theta_{Gen}} \mathcal{L}(\theta_{Gen}, \mathcal{X}^{R \to S}) = \min_{\theta_{Gen}} \frac{1}{|\mathcal{X}^{R \to S}|} \sum_{t=1}^{|\mathcal{X}^{R \to S}|} \frac{1}{|\mathcal{M}^S|} (X_t^{R \to S} - \mathcal{P}^S), \tag{12}$$

**iii) Traffic Domain Classification.** In the third step, TDA trains the discriminator model $\theta_{Dis}$ to classify the traffic data domain (⑤–⑥ shown in Fig. 5), as shown below:

$$\theta_{Dis}(X_t^{RS} \in \mathcal{X}^{RS}) = \begin{cases} P(X_t^{RS} \in \mathcal{X}^{R \to S}) \\ P(X_t^{RS} \in \mathcal{X}^S) \end{cases}, \tag{13}$$

where $\mathcal{X}^{RS} = \{X_1^{RS}, X_2^{RS}, \ldots\}$ is the traffic data mixed with the transformed data $\mathcal{X}^{R \to S}$ of the source city and the traffic data $\mathcal{X}^S$ of the target city. Besides, discriminator model $\theta_{Dis}$ is a MLP model. Then, the training process of $\theta_{Dis}$ is shown below:

$$\min_{\theta_{Dis}} \mathcal{L}(\theta_{Dis}, \mathcal{X}^{RS}) = \min_{\theta_{Dis}} \frac{1}{|\mathcal{X}^{RS}|} \sum_{t=1}^{|\mathcal{X}^{RS}|} \begin{cases} -log(P(X_t^{RS} \in \mathcal{X}^{R \to S})), & if \ X_t^{RS} \in \mathcal{X}^{R \to S} \\ -log(P(X_t^{RS} \in \mathcal{X}^S)) & , if \ X_t^{RS} \in \mathcal{X}^S \end{cases} \tag{14}$$

Next, we update the training process of the generator model $\theta_{Gen}$ in Eq. 12, as shown below:

$$\min_{\theta_{Gen}} \mathcal{L}(\theta_{Gen}, \theta_{Dis}, \mathcal{X}^{R \to S}, \mathcal{X}^{RS}) = \min_{\theta_{Gen}} \mathcal{L}(\theta_{Gen}, \mathcal{X}^{R \to S}) - \lambda_1 \mathcal{L}(\theta_{Dis}, \mathcal{X}^{RS}), \tag{15}$$

where $\lambda_1$ is the hyperparameter to control the trade-off between generator loss and discriminator loss.

## 4.3 TRAFFIC SECRET AGGREGATION

**Design Motivation.** Existing works upload gradients or models for aggregation in FTT, where attackers derive the traffic data through inference attacks (Gao et al., 2024; Wang et al., 2024b; Zheng et al., 2024). Although techniques such as Homomorphic Encryption (HE) (Rivest et al., 1978) and Differential Privacy (DP) (Dwork et al., 2006) can be employed for secure aggregation, they come with notable trade-offs. Specifically, HE introduces significant computational and communication overheads, reducing training efficiency, while DP reduces the data utility, leading to lower model accuracy. In contrast, we design the Traffic Secret Aggregation (TSA) protocol that securely transmits and aggregates the transformed data from source cities to protect traffic data privacy without sacrificing the training efficiency or model accuracy, as shown in Fig. 5 (③–④).

Specifically, it first masks the $r$-th transformed data $R_i \ X_{(r)}^{R_i \to S}$ in the client $c_i$, as shown below:

$$X_{(r)}^{(\mathcal{R} \to S, \ R_i)} = \overline{X}_{(r-1)}^{\mathcal{R} \to S} + \frac{X_{(r)}^{R_i \to S} - X_{(r-1)}^{R_i \to S}}{n}, \tag{16}$$

where $\overline{X}_{(r)}^{\mathcal{R} \to S}$ is $r$-th aggregated data. Besides, $X_{(r)}^{(\mathcal{R} \to S, \ R_i)}$ is the $r$-th mask data computed in the client $c_i$ and transmitted to the server. Note that, when $r = 0$, the client uses HE to encrypt its

transformed data and transmitted the encrypted data to the server for initial aggregation. Then, the server computes the sum of mask data from all source cities, as shown below:

$$
\begin{aligned}
\sum_{i=1}^{n} X_{(r)}^{(\mathcal{R} \to S, R_i)} &= n * \overline{X}_{(r-1)}^{\mathcal{R} \to S} + \frac{1}{n} * \sum_{i=1}^{n} X_{(r)}^{R_i \to S} - \frac{1}{n} * \sum_{i=1}^{n} X_{(r-1)}^{R_i \to S} \\
&= n * \overline{X}_{(r-1)}^{\mathcal{R} \to S} + \overline{X}_{(r)}^{\mathcal{R} \to S} - \overline{X}_{(r-1)}^{\mathcal{R} \to S} \\
&= (n-1) * \overline{X}_{(r-1)}^{\mathcal{R} \to S} + \overline{X}_{(r)}^{\mathcal{R} \to S}
\end{aligned}
\tag{17}
$$

Finally, the server gets the $r$-th aggregated data using the previous aggregated data, as shown below:

$$
\overline{\mathcal{X}}_{(r)}^{\mathcal{R} \to S} = \sum_{i=1}^{n} \mathcal{X}_{(r)}^{(\mathcal{R} \to S, R_i)} - (n-1) * \overline{\mathcal{X}}_{(r-1)}^{\mathcal{R} \to S}
\tag{18}
$$

In this way, it ensures that only the aggregated data can be accessed without revealing the individual transformed data. Besides, the client $c_i$ can train a local discriminator model $\theta_{Dis}^{R_i}$ to classify the aggregated data and individual transformed data (⑦–⑧ shown in Fig. 5), as shown below:

$$
\theta_{Dis}^{R_i}(X_t^{R_i S} \in \mathcal{X}^{R_i S}) = \begin{cases} P(X_t^{R_i S} \in \mathcal{X}^{R_i \to S}) \\ P(X_t^{R_i S} \in \overline{\mathcal{X}}^{\mathcal{R} \to S}) \end{cases},
\tag{19}
$$

where $\mathcal{X}^{R_i S} = \{X_1^{R_i S}, X_2^{R_i S}, \ldots\}$ is the traffic data mixed with the aggregated data $\overline{\mathcal{X}}^{\mathcal{R} \to S}$ and transformed data $\mathcal{X}^{R_i \to S}$. Besides, $\theta_{Dis}^{R_i}$ is a MLP model and its training process is shown below:

$$
\min_{\theta_{Dis}^{R_i}} \mathcal{L}(\theta_{Dis}^{R_i}, \mathcal{X}^{R_i S}) = \min_{\theta_{Dis}^{R_i}} \frac{1}{|\mathcal{X}^{R_i S}|} \sum_{t=1}^{|\mathcal{X}^{R_i S}|} \begin{cases} -log(P(X_t^{R_i S} \in \mathcal{X}^{R_i \to S})), & if\ X_t^{R_i S} \in \mathcal{X}^{R_i \to S} \\ -log(P(X_t^{R_i S} \in \mathcal{X}^{\mathcal{R} \to S})), & if\ X_t^{R_i S} \in \mathcal{X}^{\mathcal{R} \to S} \end{cases}
\tag{20}
$$

Therefore, given the traffic data $\mathcal{X}^{\mathcal{R} S} = \{X_1^{\mathcal{R} S}, X_2^{\mathcal{R} S}, \ldots\}$ consisting of aggregated data $\overline{\mathcal{X}}^{\mathcal{R} \to S}$ and traffic data $\mathcal{X}^S$, the updated training process of the generator model $\theta_{Gen}$ in Eq. 15 is shown below:

$$
\min_{\theta_{Gen}^{R_i}} \mathcal{L}(\theta_{Gen}^{R_i}, \mathcal{X}^{R_i \to S}) - \lambda_1 \mathcal{L}(\theta_{Dis}, \mathcal{X}^{\mathcal{R} S}) - \lambda_2 \mathcal{L}(\theta_{Dis}^{R_i}, \mathcal{X}^{R_i S}),
\tag{21}
$$

where $\theta_{Gen}^{R_i}$ and $\theta_{Dis}$ are the local generator model and global discriminator model in client $c_i$ and server $s$, respectively. Here, $\lambda_1$ and $\lambda_2$ are the hyperparameter to control the trade-off between generator loss and discriminator loss.

To further improve training efficiency and reduce communication overhead in FedTT, we incorporate a lightweight parallelization strategy termed Federated Parallel Training (FPT). The core idea is to decouple the optimization of different modules and execute them in parallel where dependencies allow. Specifically, the TVI, TDA, and TSA components operate on different representations during each communication round; therefore, their local updates can be computed concurrently without affecting correctness. This design significantly reduces wall-clock training time and lowers the frequency of communication. The detailed **algorithmic description of FPT**, overall training process, theoretical privacy analysis, and convergence analysis of FedTT are shown in **Appendix A**.

## 5 EXPERIMENT

Table 2: Statistics of evaluated datasets

| Dataset | # instances | # sensors | Interval | City | Missing Rate |
|---|---|---|---|---|---|
| PeMSD4 (PeMS, 2024) | 16992 | 307 | 5 min | San Francisco | 16.35% |
| PeMSD8 (PeMS, 2024) | 17856 | 170 | 5 min | San Bernardino | 20.09% |
| FT-AED (Coursey et al., 2024) | 1920 | 196 | 5 min | Nashville | 4.59% |
| HK-Traffic (HK, 2024) | 17856 | 411 | 5 min | Hong Kong | 13.01% |

**Datasets.** We use four traffic datasets to evaluate the proposed framework in experiments, which are widely used in traffic prediction tasks (Zhao et al., 2023; Jiang et al., 2023), as shown in Table 2. Specifically, PeMSD4 (**P4**), PeMSD8 (**P8**), FT-AED (**FT**), and HK-Traffic (**HK**) were collected in the San Francisco, San Bernardino, Nashville, and Hong Kong, respectively. Among them, three datasets are considered as three source cities, and one dataset serves as the target city, leading to four scenarios: (P8, FT, HK) → P4, (P4, FT, HK) → P8, (P4, P8, HK) → FT, and (P4, P8, FT) → HK. Besides, we select traffic flow, speed, and occupancy prediction tasks for experiments, which are also

Table 3: The overall performance comparison between different methods

| Metric | Method | (P8, FT, HK) → P4[1] | | | (P4, FT, HK) → P8 | | | (P4, P8, HK) → FT | | | (P4, P8, FT) → HK | | |
|---|---|---|---|---|---|---|---|---|---|---|---|---|---|
| | | flow | speed | occ | flow | speed | occ | flow | speed | occ | flow | speed | occ |
| MAE | 2MGTCN | 20.34 | 1.27 | 0.0077 | 16.39 | 1.09 | 0.0069 | 13.86 | 4.77 | 0.0355 | 8.49 | 1.38 | 0.0094 |
| | pFedCTP | 21.24 | 1.52 | 0.0079 | 17.06 | 1.22 | 0.0072 | 13.92 | 5.78 | 0.0415 | 9.22 | 1.22 | 0.0102 |
| | T-ISTGNN | 27.24 | 2.03 | 0.0219 | 22.75 | 1.84 | 0.0235 | 20.83 | 9.69 | 0.0571 | 9.98 | 4.24 | 0.0121 |
| | TPB | 21.06 | 1.28 | 0.0134 | 17.11 | 1.12 | 0.0081 | 13.03 | 3.59 | 0.0276 | 8.36 | 1.52 | 0.0092 |
| | ST-GFSL | 23.05 | 1.47 | 0.0161 | 19.86 | 1.47 | 0.0159 | 18.00 | 5.25 | 0.0385 | 8.42 | 2.03 | 0.0101 |
| | DastNet | 26.89 | 1.54 | 0.0165 | 19.58 | 1.41 | 0.0134 | 15.44 | 4.62 | 0.0421 | 9.09 | 3.85 | 0.0135 |
| | CityTrans | 23.94 | 1.38 | 0.0119 | 18.51 | 1.18 | 0.0108 | 13.06 | 3.60 | 0.0359 | 8.78 | 1.84 | 0.0116 |
| | TransGTR | 24.32 | 1.39 | 0.0135 | 19.53 | 1.18 | 0.0089 | 13.27 | 4.80 | 0.0337 | 9.09 | 3.92 | 0.0102 |
| | MGAT | 24.78 | 1.58 | 0.0195 | 20.16 | 1.67 | 0.0160 | 20.08 | 8.00 | 0.0469 | 9.14 | 2.88 | 0.0101 |
| | FedTT | 16.69 | 1.03 | 0.0061 | 14.11 | 0.94 | 0.0059 | 12.10 | 3.24 | 0.0249 | 7.42 | 1.05 | 0.0087 |
| RMSE | 2MGTCN | 31.61 | 2.27 | 0.0179 | 25.95 | 2.18 | 0.0131 | 17.03 | 7.49 | 0.0644 | 12.11 | 3.25 | 0.00167 |
| | pFedCTP | 33.03 | 3.12 | 0.0188 | 26.19 | 2.62 | 0.0164 | 19.94 | 9.84 | 0.0756 | 13.31 | 2.62 | 0.0212 |
| | T-ISTGNN | 35.95 | 4.14 | 0.0281 | 31.10 | 3.37 | 0.0305 | 29.42 | 13.17 | 0.1127 | 15.68 | 6.31 | 0.0230 |
| | TPB | 31.75 | 2.31 | 0.0201 | 26.35 | 2.19 | 0.0126 | 16.34 | 6.07 | 0.0493 | 11.89 | 2.98 | 0.0152 |
| | ST-GFSL | 33.65 | 3.29 | 0.0237 | 30.66 | 3.12 | 0.0260 | 22.10 | 9.69 | 0.0652 | 12.89 | 4.73 | 0.0156 |
| | DastNet | 34.96 | 3.41 | 0.0274 | 27.45 | 3.10 | 0.0299 | 22.64 | 9.72 | 0.0691 | 13.63 | 5.82 | 0.0236 |
| | CityTrans | 32.04 | 2.46 | 0.0237 | 27.91 | 2.20 | 0.0226 | 18.86 | 9.82 | 0.0514 | 13.45 | 4.72 | 0.0212 |
| | TransGTR | 33.66 | 2.43 | 0.0198 | 26.41 | 2.27 | 0.0147 | 17.11 | 7.96 | 0.0579 | 12.23 | 6.77 | 0.0180 |
| | MGAT | 32.85 | 3.43 | 0.0283 | 30.77 | 3.20 | 0.0262 | 24.62 | 11.05 | 0.1028 | 12.03 | 5.11 | 0.0162 |
| | FedTT | 27.48 | 1.93 | 0.0166 | 24.29 | 1.94 | 0.0099 | 15.91 | 5.50 | 0.0372 | 8.57 | 2.40 | 0.0145 |

[1] P4, P8, FT, and HK denote PeMSD4, PeMSD8, FT-AED, and HK-Traffic datasets, respectively.

widely studied in the community (Jiang et al., 2023; Ji et al., 2023). In addition, we report the rate of missing traffic data in these datasets, which reveals varying levels of traffic data quality issues.

**Baselines.** We compare FedTT with (i) three SOTA **methods in FTT** including T-ISTGNN (Qi et al., 2023), pFedCTP (Zhang et al., 2024b), and 2MGTCN (Yuan et al., 2025), (ii) three SOTA **Multi-Source Traffic Knowledge Transfer methods (MTT)** extended for the FTT problem including TPB (Liu et al., 2023), ST-GFSL (Lu et al., 2022), and DastNet (Tang et al., 2022), (iii) three SOTA **Single-Source Traffic Knowledge Transfer methods (STT)** for the FTT problem including CityTrans (Ouyang et al., 2024), TransGTR (Jin et al., 2023), and MGAT (Mo & Gong, 2023). In addition, we replace the TVI module of FedTT with three SOTA data imputation methods (LATC (Chen et al., 2024b), GCASTN (Peng et al., 2023), and Nuhuo (Yuan et al., 2024)) to evaluate its effects. More details about these baselines are provided in **Appendix B.1**.

**Evaluation Metrics.** We use Mean Absolute Error (MAE), Root Mean Square Error (RMSE), communication size (GB), and running time (minutes) to evaluate the utility in experiments. Besides, Mean Square Error (MSE) and Pearson Correlation Coefficient (PCC) between the reconstructed data and the ground truth data to measure the privacy-preserving ability of different methods.

The implementation details are provided in **Appendix B.2**.

## 5.1 OVERALL PERFORMANCE

To show the overall performance of different methods on traffic flow, speed, and occupancy ("occ" for short) predictions tasks, we take 60 minutes (12-time steps) of historical data as input and output the traffic prediction in the next 15 minutes (3-time steps), as shown in Table 3, where the best results are shown in blue. Here, the DyHSL (Zhao et al., 2023) model is implemented in FedTT as it achieves the state-of-the-art performance in the centralized traffic model.

As observed, the proposed FedTT framework achieves the best performance on different traffic datasets and traffic prediction tasks compared to other methods, showing its effectiveness of traffic knowledge transfer in the FTT problem, i.e., the gains range from **5.43% to 75.24%** in MAE and **2.63% to 67.54%** in RMSE. Specifically, we observe that methods originally designed for centralized multi-source transfer (e.g., TPB, DastNet) or single-source transfer (e.g., CityTrans, MGAT) suffer significant performance degradation when naively adapted to the federated setting. In contrast, recent FTT-specific methods like 2MGTCN and pFedCTP perform better but still fall short of FedTT. In contrast, FedTT's special design—traffic view imputation (TVI), domain adaptation (TDA), and secret aggregation (TSA)—effectively mitigates data quality issues, aligns heterogeneous traffic distributions, and preserves privacy, thereby enabling more accurate and robust predictions.

## 5.2 ABLATION STUDY

Fig. 6 shows the ablation study, where we removed the module of FedTT one at a time, namely FedTT without TVI (w/o TVI), FedTT without TDA (w/o TDA), and FedTT without TSA (w/o TSA). First, when TVI is absent, MAE increases by **1.49% to 9.23%**, underscoring its pivotal role as an effective way to complete the missing data. Besides, the training of TVI is completed before the FedTT's

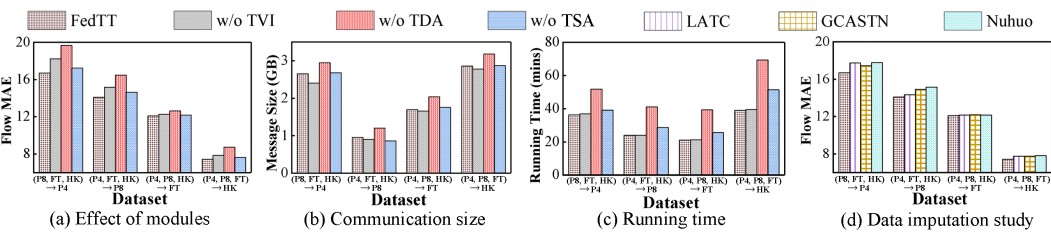

Figure 6: Ablation study of FedTT

training as it only needs to be conducted within each source city, thus not increasing communication overhead or running time during FedTT's training. Additionally, compared to other data imputation methods (i.e., LATC, GCASTN, and Nuhuo), FedTT with TVI achieves better performance, showing its effectiveness in the traffic data completion. Second, when TDA is removed, MAE increases by **4.46% to 17.86%**, which demonstrates its effectiveness in addressing traffic data distribution differences. Besides, communication overhead and running time of FedTT slightly increase compared to w/o TDA. Third, MAE of FedTT decreases **0.66% to 3.76%** compared to w/o TSA as TSA uses the averaged source data, which reduces the influence of source city's traffic patterns on the target city's model training. Besides, the communication overhead and running time of FedTT compared to w/o TSA do not change as TSA is a lightweight module for federated secure aggregation.

## 5.3 PRIVACY PROTECTION STUDY

To evaluate the privacy-preserving capabilities, we conduct the data reconstruction attack to different methods across datasets on traffic flow prediction using MSE and PCC, as illustrated in Fig. 7. As observed, FedTT demonstrates robust resistance to the data reconstruction attack, achieving a high MSE and maintaining a PCC within **2.17% to 8.81%**, not exceeding 10%, while other methods exhibit

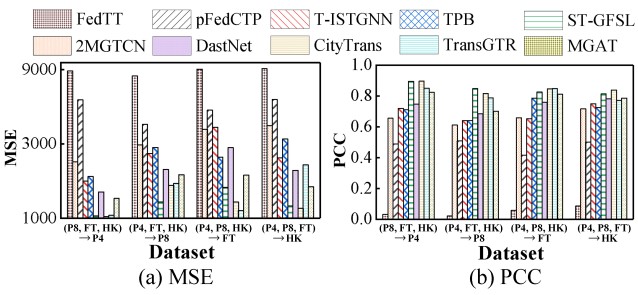

Figure 7: Privacy protection study

weaker defenses, with a lower MSE and PCC larger than 40%. These findings underscore the superiority and effectiveness of privacy protection provided by the proposed FedTT framework in FTT and highlight the limitations of privacy preservation mechanisms based solely on traditional federated learning frameworks.

## 5.4 ADDITIONAL EXPERIMENTS

- **Model Adaptability (Appendix B.3)**: To verify whether FedTT can generalize across different traffic models, we extend multiple centralized traffic models to FedTT and compare with mainstream two-stage FTT approaches.

- **Long-term Prediction (Appendix B.4)**: To assess FedTT's ability to capture long-range temporal dependencies, we evaluate traffic flow and speed prediction over the next 12 time steps.

- **Model Scalability (Appendix B.5)**: To test robustness under varying numbers of clients and data scales, we analyze FedTT's performance in diverse settings.

- **Efficiency Study (Appendix B.6)**: To evaluate the efficiency of FedTT, we analyze training time and communication overhead across different datasets.

- **Hyperparameter Sensitivity (Appendix B.7)**: To assess the stability of FedTT, we conduct sensitivity analysis over key hyperparameters.

- **Case Study (Appendix B.8)**: To evaluate FedTT's practical applicability, we conduct a real-world deployment study.

## 6 CONCLUSION

We propose FedTT, a privacy-aware federated framework for cross-city traffic knowledge transfer. FedTT proposes traffic view imputation to enhance data quality, traffic domain adapter to align cross-city distributions, and traffic secret aggregation to safeguard privacy. Extensive experiments show its superiority over baselines. **Limitations and future directions** are discussed in **Appendix C**.

ETHICS STATEMENT

We affirm that this work fully adheres to the ICLR Code of Ethics. All experiments were conducted using publicly available traffic datasets. Our work does not involve human subjects.

REPRODUCIBILITY STATEMENT

We are committed to ensuring the reproducibility of our work. All source code and the datasets used in our experiments have been made publicly available in an anonymous repository at `https://anonymous.4open.science/r/FedTT`, allowing for direct replication of our results. Besides, Section 4 of the main paper provides a comprehensive description of our experimental setup, including details on the datasets, baseline methods, evaluation metrics, and implementation specifics such as hardware configuration, data splits, and model hyperparameters. Moreover, for full transparency regarding our novel algorithms, Appendix A.2 provides detailed training algorithms (Algorithms 1 and 2) and a complete theoretical privacy analysis.

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

## APPENDIX

# APPENDIX

In the subsequent sections, we present supplementary materials to provide more details of this paper, offering deeper insights and additional technical details for readers seeking further clarification. The appendix is organized as follows.

In **Section A**, we provide the additional methodology details of our proposed FedTT framework, including (i) the federated parallel training strategy, (ii) the training process with training algorithm and complexity analysis, (iii) theoretical privacy analysis, and (iv) convergence analysis.

In **Section B**, we describe the extensive experimental details to provide more information about experimental settings and further demonstrate the superior performance of the proposed FedTT framework, including (i) compared baselines, (ii) implementation details, and (iii) the details experimental results of model adaptability, efficiency, scalability, hyperparameter sensitivity, and case studies.

In **Section C**, we discuss several limitations of the proposed FedTT framework that warrant further exploration.

In **Section D**, we provide a transparent account of the role Large Language Models (LLMs) played in the preparation of this manuscript, in accordance with ethical research practices and ICLR's commitment to research integrity.

## A  METHODOLOGY DETAILS

### A.1  FEDERATED PARALLEL TRAINING

To improve the training efficiency, FedTT introduces the federated parallel training strategy to reduce the data transmission and train the models in parallel.

**i) Split Learning.** To reduce the communication overhead and improve the training efficiency, it employs split learning (Meng et al., 2021) to decompose the sequential training process into the client and server training, and freeze the data required by the client and server. Specifically, the client $c_i$ stores and freezes the data sent by the server for $\theta_{Gen}^{R_i}$ and $\theta_{Dis}^{R_i}$ training in Eqs. 21 and 20, respectively:

$$\min_{\theta_{Gen}^{R_i}} \mathcal{L}(\theta_{Gen}^{R_i}, \mathcal{X}^{R_i \to S}) - \lambda_1 * Fr(\mathcal{L}(\theta_{Dis}, \mathcal{X}^{\mathcal{R}S})) - \lambda_2 \mathcal{L}(\theta_{Dis}^{R_i}, \mathcal{X}^{R_i S}), \tag{22}$$

$$\min_{\theta_{Dis}^{R_i}} \frac{1}{|\mathcal{X}^{R_i S}|} \sum_{t=1}^{|\mathcal{X}^{R_i S}|} \begin{cases} -log(P(X_t^{R_i S} \in \mathcal{X}^{R_i \to S})), if\ X_t^{R_i S} \in \mathcal{X}^{R_i \to S} \\ -log(P(X_t^{R_i S} \in \mathcal{X}^{\mathcal{R} \to S})), if\ X_t^{R_i S} \in Fr(\mathcal{X}^{\mathcal{R} \to S}) \end{cases}, \tag{23}$$

where $Fr(\cdot)$ is the frozen function and uses the historical cached data, which updates every 5 rounds. Besides, the server $s$ stores and freezes the data uploaded by the client to compute the aggregated data for $\theta_{Dis}$ and traffic model $\theta_{TP}$ training in Eqs. 14 and 3, respectively:

$$\min_{\theta_{Dis}} \frac{1}{|\mathcal{X}^{\mathcal{R}S}|} \sum_{t=1}^{|\mathcal{X}^{\mathcal{R}S}|} \begin{cases} -log(P(X_t^{\mathcal{R}S} \in \mathcal{X}^{\mathcal{R} \to S})),\ if\ X_t^{\mathcal{R}S} \in Fr(\mathcal{X}^{\mathcal{R} \to S}) \\ -log(P(X_t^{\mathcal{R}S} \in \mathcal{X}^{S}))\quad,\ if\ X_t^{\mathcal{R}S} \in \mathcal{X}^{S} \end{cases}, \tag{24}$$

$$\min_{\theta_{TP}} \mathcal{L}(\theta_{TP}, Fr(D^{\mathcal{R} \to S}), D^S) \tag{25}$$

**ii) Parallel Optimization.** To further improve the training parallelism, it proposes parallel optimization to reduce data dependencies on the client and server. Specifically, the client $c_i$ caches and freezes the local data for $\theta_{Gen}^{R_i}$ and $\theta_{Dis}^{R_i}$ parallel training in Eqs 22 and 23, as shown below:

$$\min_{\theta_{Gen}^{R_i}} \mathcal{L}(\theta_{Gen}^{R_i}, \mathcal{X}^{R_i \to S}) - \lambda_1 * Fr(\mathcal{L}(\theta_{Dis}, \mathcal{X}^{\mathcal{R}S})) - \lambda_2 * Fr^{'}(\mathcal{L}(\theta_{Dis}^{R_i}, \mathcal{X}^{R_i S})), \tag{26}$$

$$\min_{\theta_{Dis}^{R_i}} \frac{1}{|\mathcal{X}^{R_i S}|} \sum_{t=1}^{|\mathcal{X}^{R_i S}|} \begin{cases} -log(P(X_t^{R_i S} \in \mathcal{X}^{R_i \to S})), if\ X_t^{R_i S} \in Fr^{'}(\mathcal{X}^{R_i \to S}) \\ -log(P(X_t^{R_i S} \in \mathcal{X}^{\mathcal{R} \to S}))\ , if\ X_t^{R_i S} \in Fr(\mathcal{X}^{\mathcal{R} \to S}) \end{cases}, \tag{27}$$

where $Fr^{'}(\cdot)$ is the frozen function and uses the historical cached data, which updates each round.

---

**Algorithm 1:** The training of the FedTT framework in the client $c_i$

---

**Input:** Server $s$ (target city $S$); local data $\mathcal{X}^{R_i}$; models $\theta_{\text{TVI}}, \theta_{\text{Gen}}^{R_i}, \theta_{\text{Dis}}^{R_i}$

1   $\widetilde{\mathcal{X}}^{R_i} \leftarrow \text{COMPLETE}(\theta_{\text{TVI}}, \mathcal{X}^{R_i})$;         `// Impute missing data`

2   **for** $tr \leftarrow 1$ **to** $T$ **do**

3     **for** *each* $X_{(r)}^{R_i} \in \widetilde{\mathcal{X}}^{R_i}$ **do**

4       $X_{(r)}^{R_i \to S} \leftarrow \text{TRANSFORM}(\theta_{\text{Gen}}^{R_i}, X_{(r)}^{R_i})$;       `// Domain transform`

5       $\text{CLASSIFY}(\theta_{\text{Dis}}^{R_i}, X_{(r)}^{R_i \to S})$;

6       **if** $tr = 1 \wedge r = 1$ **then**

7         $E_{(r)}^{R_i \to S} \leftarrow \text{ENCRYPT}(X_{(r)}^{R_i \to S})$;

8         $\text{SEND}(s, E_{(r)}^{R_i \to S})$;

9       **else**

10         **if** $tr = 1 \wedge r = 2$ **then**

11           $\overline{E}_{(r-1)}^{\mathcal{R} \to S} \leftarrow \text{GET}(s, r)$;

12           $\overline{X}_{(r-1)}^{\mathcal{R} \to S} \leftarrow \text{DECRYPT}(\overline{E}_{(r-1)}^{\mathcal{R} \to S})$;

13         **else**

14           $\overline{X}_{(r-1)}^{\mathcal{R} \to S} \leftarrow \text{GET}(s, r)$;

15         $\text{CLASSIFY}(\theta_{\text{Dis}}^{R_i}, \overline{X}_{(r-1)}^{\mathcal{R} \to S})$;

16         $X_{(r)}^{(\mathcal{R} \to S, R_i)} \leftarrow \overline{X}_{(r-1)}^{\mathcal{R} \to S} + X_{(r)}^{R_i \to S} - X_{(r-1)}^{R_i \to S}$;      `// Masking`

17         $\text{SEND}(s, X_{(r)}^{(\mathcal{R} \to S, R_i)})$;

---

### A.2   TRAINING PROCESS

Before the training of the FedTT framework, clients (i.e., source cities) train the spatial view expansion model $\theta_{SV}$ and the temporal view expansion model $\theta_{TV}$ in the TVI module $\theta_{TVI}$ by minimizing the loss in Eqs. 7 and 9, as shown below:

$$\min_{\theta_{TVI}} \mathcal{L}(\theta_{TVI}, \mathcal{V}_{SV}, \mathcal{V}_{TV}) = \min_{\theta_{SV}} \mathcal{L}(\theta_{SV}, \mathcal{V}_{SV}) + \min_{\theta_{TV}} \mathcal{L}(\theta_{TV}, \mathcal{V}_{TV}), \tag{28}$$

where $\mathcal{V}_{SV}$ and $\mathcal{V}_{TV}$ are the set of traffic subviews at different times obtained by spatial view extension and temporal view enhancement, respectively. During the training of the FedTT framework, the client $c_i$ trains the local generator model $\theta_{Gen}^{R_i}$ and the local discriminator model $\theta_{Dis}^{R_i}$ by minimizing the loss in Eqs. 20 and 21, as shown below:

$$\min_{\theta_{Gen}^{R_i}} \mathcal{L}(\theta_{Gen}^{R_i}, \theta_{Dis}, \theta_{Dis}^{R_i}, \mathcal{X}^{R_i \to S}, \mathcal{X}^{\mathcal{R}S}, \mathcal{X}^{R_i S}) + \min_{\theta_{Dis}^{R_i}} \mathcal{L}(\theta_{Dis}^{R_i}, \mathcal{X}^{R_i S}), \tag{29}$$

where $\mathcal{X}^{\mathcal{R}S}$ is the traffic data consisting of the aggregated data $\overline{\mathcal{X}}^{\mathcal{R} \to S}$ and traffic data $\mathcal{X}^S$ of the target city $S$, and $\mathcal{X}^{R_i S}$ is the traffic data consisting of the aggregated data $\overline{\mathcal{X}}^{\mathcal{R} \to S}$ and transformed data $\mathcal{X}^{R_i \to S}$ of the source city $R_i$. Besides, the server $s$ trains the global discriminator model $\theta_{Dis}$ and traffic model $\theta_{TP}$ by minimizing the loss in Eqs. 14 and 3, as shown below:

$$\min_{\theta_{Dis}} \mathcal{L}(\theta_{Dis}, \mathcal{X}^{\mathcal{R}S}) + \min_{\theta_{TP}} \mathcal{L}(\theta_{TP}, \overline{D}^{\mathcal{R} \to S}, D^S), \tag{30}$$

where $\overline{D}^{\mathcal{R} \to S}$ is the aggregated traffic dataset whose traffic domain is transformed from source cities to the target city $S$, and $D^S$ is the traffic dataset of the target city $S$.

---

**Algorithm 2:** The training of the FedTT framework in the server $s$

---

**Input:** Clients $\mathcal{C} = \{c_1, \ldots, c_n\}$ with source cities $\mathcal{R} = \{R_1, \ldots, R_n\}$; models $\theta_{\text{Dis}}, \theta_{\text{TP}}$; local data $\mathcal{X}^S$

1   **for** $tr \leftarrow 1$ **to** $T$ **do**

2     **for** $r \leftarrow 1$ **to** $R$ **do**

3       **if** $tr = 1 \wedge r = 1$ **then**

4         $\{E_{(r)}^{R_1 \to S}, E_{(r)}^{R_2 \to S}, \ldots\} \leftarrow \text{GET}(\mathcal{C}, r)$;     // Receive encrypted data

5         $\overline{E}_{(r)}^{\mathcal{R} \to S} \leftarrow \sum_{i=1}^{n} E_{(r)}^{R_i \to S}$;        // Aggregate encrypted data

6         $\text{SEND}(\mathcal{C}, \overline{E}_{(r)}^{\mathcal{R} \to S})$;

7       **else**

8         $\{X_{(r)}^{(\mathcal{R} \to S, R_1)}, X_{(r)}^{(\mathcal{R} \to S, R_2)}, \ldots\} \leftarrow \text{GET}(\mathcal{C}, r)$;    // Receive masked data

9         $\overline{X}_{(r)}^{\mathcal{R} \to S} \leftarrow \sum_{i=1}^{n} X_{(r)}^{(\mathcal{R} \to S, R_i)} - (n-1) \cdot \overline{X}_{(r-1)}^{\mathcal{R} \to S}$;    // Aggregate masked data

10         $\text{CLASSIFY}(\theta_{\text{Dis}}, \overline{X}_{(r)}^{\mathcal{R} \to S})$;

11         $\text{SEND}(\mathcal{C}, \overline{X}_{(r)}^{\mathcal{R} \to S})$;

12    $\text{CLASSIFY}(\theta_{\text{Dis}}, \mathcal{X}^S)$;         // Update on server local data

13    $\text{PREDICTION}(\theta_{\text{TP}}, \overline{\mathcal{X}}^{\mathcal{R} \to S}, \mathcal{X}^S)$;         // Traffic prediction

---

**Training Algorithm**. For convenient method reproduction, we provide detailed training Algorithms 1 and 2 of the FedTT framework, including the client and server.

In the client (i.e., Algorithm 1), the target city acts as the server. Before the training process, the client completes the missing traffic data through the traffic view imputation method (line 1). During each training round and each traffic data (lines 2–3), it first transforms the data from the traffic domain of the source city to that of the target city using the local generator model (line 4) and classifies the transformed data using the local discriminator model (line 5). If the training process is in the first round using the first data instance (line 6), the client encrypts the transformed data using homomorphic encryption and sends it to the server (lines 7-8). Otherwise, if the training process is in the first round using the second data instance (lines 9-10), the client gets the encrypted data and decrypts it to get the previous aggregated data (lines 11-12). For subsequent rounds or data instance, the client directly gets the previous aggregated data from the server without decryption (lines 13-14). In either case, it classifies the previous aggregated data using its local discriminator model (line 15). Then it masks the transformed data using the previous aggregated and transformed data (line 16). Finally, it sends the mask data to the server for data aggregation (lines 17).

In the server (i.e., Algorithm 2), the source cities act as the clients. During each training round and each traffic data (lines 1–2), if the training process is in the first round using the first data instance (line 3), the server gets the encrypted data from clients (line 4). Then, it aggregates them by summing up, and send the aggregated encrypted data to back to the clients for further processing (lines 5-6). For subsequent rounds or data instances (line 7), the server gets the mask data from clients (line 8). Then, it aggregates the masked data using the previous aggregated data (line 9). Next, it classifies the aggregated data using its global discriminator model and sends the aggregated data back to the clients (lines 10–11). Finally, at the end of each training round, it classifies local traffic data and performs traffic prediction using the aggregated and local traffic data (lines 12–13).

**Complexity Analysis**. We also give the complete complexity analysis for the training of the FedTT framework, i.e., Algorithms 1 and 2. For the client (i.e., Algorithm 1), the training complexity is $O((|\mathcal{M}^{R_i}| + |\mathcal{M}^S|) \times (F_1 \times H)^2 \times |\mathcal{X}^{R_i}|)$ at each round. For the server (i.e., Algorithm 2), the training complexity is $O((|\mathcal{M}^S| \times (F_1 \times H)^2 + MC(\theta_{TP})) \times (|\mathcal{X}^S| + \sum_{i=1}^{n} |\mathcal{X}^{R_i}|))$ at each round. Here, $|\mathcal{M}^{R_i}|$ and $|\mathcal{M}^S|$ are the number of sensors in the source city $R_i$ and target city $S$, respectively. Besides, $|\mathcal{X}^{R_i}|$ and $|\mathcal{X}^S|$ are the number of traffic data in the source city $R_i$ and target city $S$, respectively. In addition, $F_1 = 3$ is the dimensions of traffic data features, and $H = 1024$ is the hidden dimensions of the three-layer MLP model in $\theta_{Gen}^{R_i}$ and $\theta_{Dis}^{R_i}$. Moreover, $MC(\theta_{TP})$ is the model complexity of $\theta_{TP}$ (i.e., $\theta_{DyHSL}$).

### A.3 THEORETICAL PRIVACY ANALYSIS

The privacy protection mechanism of the proposed FedTT framework comprises two stages. First, it uses the Traffic Domain Adapter (TDA) to transform the data from the traffic domain of source cities to that of the target city, where the parameters of the TDA model are private and not shared with the server and other clients. Second, it performs Traffic Secret Aggregation (TSA) to secure mask and aggregate the transformed data. Consequently, an attacker must first reverse-engineer the transformed data from the aggregated data and then infer the original traffic data from the transformed data. To rigorously analyze the privacy-preserving capability of these two stages, we first define the threat model as follows.

**Threat Model**. Following previous works (Zhang et al., 2024a; Tong et al., 2025; Zhao et al., 2024) in federated learning scenarios, we assume that the server acts as a semi-honest adversary who will honestly execute the required operations (e.g., aggregation) but also remains curious about the private data in clients. In the FTT problem, the server may perform inference attacks to infer the raw instance-level traffic data of clients based on the adversary knowledge, including the client model architecture, privacy-preserving mechanism, and the intermediate data (e.g., model parameters or training gradients) uploaded by clients.

Based on this, we analyze the privacy leakage of FedTT using mutual information (Kreer, 1957) as follows.

**Privacy Protection in Traffic Domain Adapter.** Given the transformed data $\mathcal{X}^{R_i \to S}$ of the source city $R_i$, the attacker aims to infer the original traffic data $\mathcal{X}^{R_i}$, where $\mathcal{X}^{R_i \to S}$ is derived from $\mathcal{X}^{R_i}$ in Eq.10 as shown below:

$$\mathcal{X}^{R_i} \xrightarrow{\theta_{TDA}} \mathcal{X}^{R_i \to S}, \tag{31}$$

where the TDA model $\theta_{TDA}$ is private and inaccessible. Since this process represents a deterministic mapping, the privacy leakage can be quantified as:

$$I(\mathcal{X}^{R_i}; \mathcal{X}^{R_i \to S}) = H(\mathcal{X}^{R_i \to S}) - H(\mathcal{X}^{R_i \to S} | \mathcal{X}^{R_i}) = H(\mathcal{X}^{R_i \to S}), \tag{32}$$

where $H(\cdot)$ denotes entropy and $H(\mathcal{X}^{R_i \to S} | \mathcal{X}^{R_i}) = 0$ due to the nature of deterministic mapping. Since $\mathcal{X}^{R_i \to S}$ is derived from $\mathcal{X}^{R_i}$ through the private TDA model $\theta_{TDA}$, the amount of privacy leakage can be further expressed as follows:

$$
\begin{aligned}
I(\mathcal{X}^{R_i}; \mathcal{X}^{R_i \to S}) &\leq I(\mathcal{X}^{R_i}; \mathcal{X}^{R_i \to S}, \theta_{TDA}) \\
&= I(\mathcal{X}^{R_i}; \theta_{TDA}) + I(\mathcal{X}^{R_i}; \mathcal{X}^{R_i \to S} | \theta_{TDA}) \\
&= H(\mathcal{X}^{R_i \to S} | \theta_{TDA}) \propto \frac{|\mathcal{M}^{R_i}|}{|\theta_{TDA}| * |\mathcal{M}^S|},
\end{aligned}
\tag{33}
$$

where $|\theta_{TDA}|$ is the parameter space of the TDA model. As $\theta_{TDA}$ aligns the distribution of $\mathcal{X}^{R_i \to S}$) to the traffic domain of the target city through traffic domain alignment, reducing its correlation with source city's traffic domain, $H(\mathcal{X}^{R_i \to S} | \theta_{TDA})$ takes on a small value, thereby minimizing the privacy leakage $I(\mathcal{X}^{R_i}, \mathcal{X}^{R_i \to S})$.

**Privacy Protection in Traffic Secure Aggregation.** Given the aggregated data $\overline{\mathcal{X}}^{\mathcal{R} \to S}$, the attacker aims to infer the transformer data $\mathcal{X}^{R_i \to S}$ of the source city $R_i$, where $\mathcal{X}^{(R_i \to S, R_i)}$ is derived from $\mathcal{X}^{R_i \to S}$ in Eq.16 as shown below:

$$\overline{\mathcal{X}}^{\mathcal{R} \to S} = \frac{1}{n}(\mathcal{X}^{R_i \to S} + \sum_{j=1 \& j \neq i}^{n} \mathcal{X}^{R_j \to S}) \tag{34}$$

Since the traffic domains of source cities are aligned to that of the target city, they are from Independent Identically Distributed (IID), and the privacy leakage can be quantified as:

$$
\begin{aligned}
I(\mathcal{X}^{R_i \to S}; \overline{\mathcal{X}}^{\mathcal{R} \to S}) &= H(\overline{\mathcal{X}}^{\mathcal{R} \to S}) - H(\overline{\mathcal{X}}^{\mathcal{R} \to S} | \mathcal{X}^{R_i \to S}) \\
&\leq H(\mathcal{X}^{R_i \to S}) - H(\frac{1}{n} \sum_{j=1 \& j \neq i}^{n} \mathcal{X}^{R_j \to S}) \\
&\leq \frac{H(\mathcal{X}^{R_i \to S})}{n} \propto \frac{1}{n * |\mathcal{M}^S|}
\end{aligned}
\tag{35}
$$

Since the above two processes is a Markov Chain (Markov, 1906), i.e., $\mathcal{X}^{R_i} \to \mathcal{X}^{R_i \to S} \to \overline{\mathcal{X}}^{\mathcal{R} \to S}$, the total amount of the privacy leakage can be bounded using the data processing inequality (Shannon, 1948):

$$I(\mathcal{X}^{R_i}; \overline{\mathcal{X}}^{\mathcal{R} \to S}) \leq min(I(\mathcal{X}^{R_i}; \mathcal{X}^{R_i \to S}), I(\mathcal{X}^{R_i \to S}; \overline{\mathcal{X}}^{\mathcal{R} \to S}))$$

$$\leq min(H(\mathcal{X}^{R_i \to S} | \theta_{TDA}), \frac{H(\mathcal{X}^{R_i \to S})}{n}) \tag{36}$$

This analysis demonstrates that the FedTT framework effectively minimizes privacy leakage by leveraging both TDA and TSA, ensuring robust privacy protection in federated traffic knowledge transfer.

## A.4 CONVERGENCE ANALYSIS

In this section, we present a convergence analysis for the Traffic Domain Adapter (TDA) used in our FedTT framework. The TDA mapping is defined as Eq. 11 and the generator is trained with the prototype alignment loss

$$L_{\text{align}}(\theta_{\text{Gen}}) = \frac{1}{|X^{R \to S}|} \sum_t \frac{1}{|M_S|} \left\| X_t^{R \to S}(\theta_{\text{Gen}}) - P_S \right\|_2^2, \tag{37}$$

while the discriminator is trained using the domain classification loss

$$L_{\text{Dis}}(\theta_{\text{Dis}}) = -\mathbb{E}_{x \in X^{R \to S}} \log D_{\theta_{\text{Dis}}}(x) - \mathbb{E}_{x \in X^S} \log(1 - D_{\theta_{\text{Dis}}}(x)) \tag{38}$$

The generator objective is given by

$$L_{\text{Gen}}(\theta_{\text{Gen}}; \theta_{\text{Dis}}) = L_{\text{align}}(\theta_{\text{Gen}}) - \lambda_1 L_{\text{Dis}}(\theta_{\text{Dis}}; X^{R \to S}(\theta_{\text{Gen}}), X^S) \tag{39}$$

Thus the complete TDA optimization problem can be written as the following min–max game:

$$\min_{\theta_{\text{Gen}}} \max_{\theta_{\text{Dis}}} \Phi(\theta_{\text{Gen}}, \theta_{\text{Dis}}) = L_{\text{align}}(\theta_{\text{Gen}}) - \lambda_1 L_{\text{Dis}}(\theta_{\text{Dis}}; X^{R \to S}(\theta_{\text{Gen}}), X^S) \tag{40}$$

**Convergence of Prototype Alignment (Non-adversarial Case).** We first analyze the simplified problem in which only the alignment loss $L_{\text{align}}$ is considered:

$$\min_{\theta_{\text{Gen}}} L_{\text{align}}(\theta_{\text{Gen}}) = \frac{1}{N} \sum_t \left\| f_{\theta_{\text{Gen}}}(X_t^R) - P_S \right\|_2^2 \tag{41}$$

**Assumptions.** We make the following standard assumptions:

1. **(Smoothness)** $L_{\text{align}}$ is differentiable and its gradient is $L$-Lipschitz:

$$\|\nabla L_{\text{align}}(\theta) - \nabla L_{\text{align}}(\theta')\| \leq L\|\theta - \theta'\| \tag{42}$$

2. **(Lower-bounded)** $L_{\text{align}}(\theta) \geq 0$.
3. **(Learning rate)** The step size satisfies $\eta \in (0, 2/L)$

**TGradient update.** The generator is updated via

$$T\theta_{\text{Gen}}^{k+1} = \theta_{\text{Gen}}^k - \eta \nabla L_{\text{align}}(\theta_{\text{Gen}}^k) \tag{43}$$

**TLemma 1 (Monotonic descent).** Under Assumptions 1—3,

$$TL_{\text{align}}(\theta^{k+1}) \leq L_{\text{align}}(\theta^k) - c\|\nabla L_{\text{align}}(\theta^k)\|_2^2, \quad c = \eta - \frac{L\eta^2}{2} > 0 \tag{44}$$

**Corollary 1 (Stationary-point convergence).** Since $L_{\text{align}}$ is nonnegative and monotonically decreasing,

$$TL_{\text{align}}(\theta^k) \to L_\infty < \infty, \qquad \|\nabla L_{\text{align}}(\theta^k)\| \to 0 \tag{45}$$

Thus the generator converges to a stationary point $\theta_{\text{Gen}}^*$ of $L_{\text{align}}$.

**Convergence of the Full Min–Max Game.** With the discriminator included, the training alternates between:

$$\theta_{\text{Gen}}^{k+1} = \theta_{\text{Gen}}^k - \eta_g \nabla_{\theta_{\text{Gen}}} L_{\text{Gen}}(\theta_{\text{Gen}}^k; \theta_{\text{Dis}}^k), \tag{46}$$

$$\theta_{\text{Dis}}^{k+1} = \theta_{\text{Dis}}^k + \eta_d \nabla_{\theta_{\text{Dis}}} L_{\text{Dis}}(\theta_{\text{Dis}}^k; \theta_{\text{Gen}}^{k+1}) \tag{47}$$

**Assumptions.** Following standard GAN convergence analyses:

1. Both $L_{\text{Gen}}$ and $L_{\text{Dis}}$ have Lipschitz-continuous gradients.
2. The game is locally convex–concave (or pseudo-convex–pseudo-concave) around a solution.
3. Learning rates $\eta_g, \eta_d$ are sufficiently small.

**Theorem 1 (Local Nash equilibrium convergence).** Under the above assumptions, the alternating gradient descent–ascent iterations converge to a local Nash equilibrium $(\theta_{\text{Gen}}^*, \theta_{\text{Dis}}^*)$, where

$$\nabla_{\theta_{\text{Gen}}} \Phi(\theta_{\text{Gen}}^*, \theta_{\text{Dis}}^*) = 0, \qquad \nabla_{\theta_{\text{Dis}}} \Phi(\theta_{\text{Gen}}^*, \theta_{\text{Dis}}^*) = 0 \tag{48}$$

At this point, the generator produces $X^{R \to S}$ that are both close to the target prototype $P_S$ and indistinguishable from real target data $X^S$ by the discriminator.

**Implication for the FedTT Framework.** Once the TDA converges, we obtain a stable mapping $X^R \xrightarrow{\theta_{\text{TDA}}^*} X^{R \to S}$ that aligns the source-domain features to a target-domain representation. Thus, the overall FedTT optimization in Eq. 3 reduces to standard federated optimization on a unified feature domain, allowing FedTT to inherit the convergence guarantees of FedAvg under standard smoothness and bounded-variance assumptions.

# B EXPERIMENTAL DETAILS

## B.1 BASELINES

We compare the FedTT framework with state-of-the-art baselines. First, we compare FedTT with three SOTA transfer methods in Federated Traffic Knowledge Transfer (FTT), including T-ISTGNN (Qi et al., 2023), pFedCTP (Zhang et al., 2024b), and 2MGTCN (Yuan et al., 2025), as detailed below.

- **T-ISTGNN.** It designs a spatio-temporal GNN-based approach with an inductive mode for cross-region traffic prediction.
- **pFedCTP.** It designs an ST-Net for privacy-preserving and cross-city traffic prediction with personalized federated learning.
- **2MGTCN.** It designs multi-modal GCNs and TCNs to capture spatial and temporal information and enhance adaptability across cities.

Besides, we compare FedTT with three SOTA transfer methods in Multi-Source Traffic Knowledge Transfer (MTT), including TPB (Liu et al., 2023), ST-GFSL (Lu et al., 2022), and DastNet (Tang et al., 2022), as detailed below.

- **TPB.** It utilizes a traffic patch encoder to create a traffic pattern bank for the cross-city few-shot traffic knowledge transfer.
- **ST-GFSL.** It transfers traffic knowledge through model parameter matching to retrieve similar spatio-temporal features.
- **DastNet.** It employs graph learning and domain adaptation to create domain-invariant node embeddings for the traffic data.

In addition, we compare FedTT with three SOTA transfer methods in Single-Source Traffic Knowledge Transfer (STT), including CityTrans (Ouyang et al., 2024), TransGTR (Jin et al., 2023), and MGAT (Mo & Gong, 2023), as detailed below.

- **CityTrans.** It proposes a domain adversarial model with knowledge transfer for spatio-temporal prediction across cities.

- **TransGTR.** It leverages adaptive spatio-temporal knowledge and domain-invariant features for TP in data-scarce cities.
- **MGAT.** It extracts multi-granular regional features from source cities to enhance the effectiveness of knowledge transfer.

Moreover, we extend three classic (Gated Recurrent Unit (GRU) (Cho et al., 2014), Convolutional Neural Network (CNN) (LeCun et al., 1989), and Multi-Layer Perceptron (MLP) (Rumelhart et al., 1986)) and the following SOTA traffic models in FedTT and the existing two-stage transfer methods in FTT (referred as FTL), including ST-SSL (Ji et al., 2023), DyHSL (Zhao et al., 2023) and PDFormer (Jiang et al., 2023), as detailed below.

- **ST-SSL.** It models traffic data at attribute and structure levels for spatial and temporal heterogeneous-aware traffic prediction.
- **DyHSL.** It leverages hypergraph structure information to extract dynamic and high-order relations of traffic road networks.
- **PDFormer.** It introduces self-attention and feature transformation for dynamic and flow-delay-aware traffic prediction.

To evaluate the Traffic View Imputation (TVI) method of FedTT in the ablation study, we replace this module with three SOTA data imputation methods, including LATC (Chen et al., 2024b), GCASTN (Peng et al., 2023), and Nuhuo (Yuan et al., 2024), as detailed below.

- **LATC.** It integrates temporal variation as a regularization term to accurately impute missing spatio-temporal traffic data.
- **GCASTN.** It uses self-supervised learning and a missing-aware attention mechanism to impute the missing traffic data.
- **Nuhuo.** It uses graph neural networks and self-supervised learning to accurately estimate missing traffic speed histograms.

### B.2 IMPLEMENTATION

All baselines run under their optimal settings. Besides, we use 5% train data, 10% validation data, and 10% test data in the target city for all methods. In addition, the MLP model used in FedTT is three-layer with the GELU (Hendrycks & Gimpel, 2016) activation and 1024 hidden dimensions. Moreover, all experiments are conducted with four nodes, one as a server and the other three nodes as clients, each equipped with two Intel Xeon CPU E5-2650 12-core processors and two NVIDIA GeForce RTX 3090 with network bandwidth of 100 MB/s. Finally, the learning rate used in our method is 0.0005 with the batch size of 128 for 1,000 training rounds. The average performance over 5 independent runs is reported, which is a common practice in prior work on traffic knowledge transfer (Ouyang et al., 2024; Lu et al., 2022), ensuring fair comparability with existing benchmarks.

To evaluate the privacy robustness, we implement a standard optimization-based reconstruction attack commonly used in inversion settings. The attacker is assumed to know the model architecture and to have access only to the exchanged information $Y$ (i.e., gradients or intermediate representations). The attack initializes dummy inputs and model parameters randomly and optimizes them by minimizing the MSE between the model outputs on the dummy inputs and the observed $Y$. The optimized dummy inputs are taken as reconstructed traffic data and compared with the private ground truth using MSE and PCC. We use the Adam optimizer with a learning rate of 1e-5, a batch size of 1, and early stopping after 500 steps without improvement.

### B.3 MODEL ADAPTABILITY

Table 4 shows the overall performance when extending existing centralized traffic models (i.e., GRU (Cho et al., 2014), CNN (LeCun et al., 1989), MLP (Rumelhart et al., 1986), ST-SSL (Ji et al., 2023), DyHSL (Zhao et al., 2023) and PDFormer (Jiang et al., 2023)) in FTT using FedTT and FTL methods with MAE, where the best results are shown in blue. As observed, all centralized traffic models extended in FedTT achieve the best performance compared to those extended in FTL, also showing its effectiveness of traffic knowledge transfer in FTT, i.e., the gains range from **5.13% to**

Table 4: The overall performance (MAE) comparison when extending centralized traffic models

| Model | Method | (P8, FT, HK) → P4 | | | (P4, FT, HK) → P8 | | | (P4, P8, HK) → FT | | | (P4, P8, FT) → HK | | |
|---|---|---|---|---|---|---|---|---|---|---|---|---|---|
| | | flow | speed | occ | flow | speed | occ | flow | speed | occ | flow | speed | occ |
| GRU | FTL[1] | 29.27 | 3.39 | 0.0282 | 23.44 | 2.40 | 0.0253 | 21.16 | 12.18 | 0.0712 | 10.11 | 4.60 | 0.0125 |
| | FedTT | 25.93 | 2.24 | 0.0220 | 20.73 | 2.21 | 0.0213 | 17.34 | 5.67 | 0.0401 | 9.33 | 2.86 | 0.0101 |
| CNN | FTL | 31.46 | 4.55 | 0.0317 | 27.60 | 3.27 | 0.0267 | 24.55 | 9.05 | 0.0803 | 9.74 | 5.92 | 0.0169 |
| | FedTT | 26.82 | 2.84 | 0.0274 | 22.20 | 2.41 | 0.0217 | 17.44 | 6.27 | 0.0472 | 9.24 | 3.92 | 0.0113 |
| MLP | FTL | 34.01 | 3.66 | 0.0276 | 30.24 | 2.88 | 0.0246 | 22.66 | 14.43 | 0.0743 | 10.87 | 5.23 | 0.0146 |
| | FedTT | 28.08 | 2.17 | 0.0250 | 23.79 | 2.40 | 0.0212 | 17.66 | 7.35 | 0.0480 | 9.68 | 3.27 | 0.0102 |
| ST-SSL | FTL | 26.76 | 2.26 | 0.0176 | 20.06 | 1.88 | 0.0226 | 19.43 | 7.78 | 0.0605 | 9.43 | 4.36 | 0.0117 |
| | FedTT | 22.28 | 1.34 | 0.0096 | 17.14 | 1.27 | 0.0114 | 13.38 | 4.88 | 0.0400 | 8.76 | 1.65 | 0.0097 |
| DyHSL | FTL | 18.61 | 1.39 | 0.0131 | 16.71 | 1.40 | 0.0144 | 16.96 | 6.04 | 0.0324 | 8.63 | 2.97 | 0.0103 |
| | FedTT | 16.69 | 1.03 | 0.0061 | 14.11 | 0.94 | 0.0059 | 12.10 | 3.24 | 0.0249 | 7.42 | 1.05 | 0.0087 |
| PDFormer | FTL | 26.99 | 2.31 | 0.0194 | 22.85 | 1.80 | 0.0232 | 17.92 | 6.57 | 0.0433 | 9.17 | 3.29 | 0.0108 |
| | FedTT | 22.05 | 1.43 | 0.0125 | 17.67 | 1.36 | 0.0127 | 13.09 | 3.53 | 0.0314 | 8.22 | 1.22 | 0.0091 |

[1] FTL refers to the two-stage method of existing methods in FTT.

**64.65%**. Note that the DyHSL model has the best performance in centralized traffic models and is implemented in FedTT as the default model in other experiments.

## B.4 LONG-TERM TRAFFIC PREDICTION

To evaluate long-term traffic prediction capabilities, we illustrate the performance of different methods over the next 60 minutes (12 time steps) for traffic flow and speed prediction using MAE metric, as shown in Fig. 8. As observed, the FedTT framework outperforms all other methods, i.e., the gains range from **5.03% to 64.41%**, showing its effectiveness of long-term traffic prediction in federated traffic transfer knowledge. Therefore, the

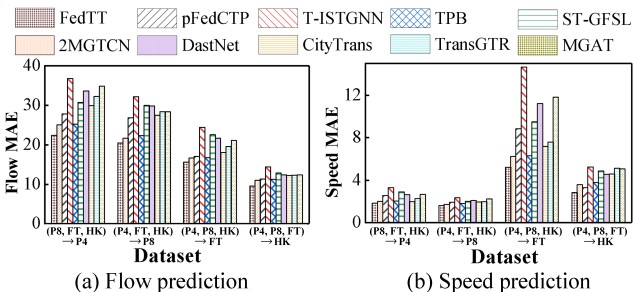

(a) Flow prediction      (b) Speed prediction

Figure 8: Long-term traffic prediction

proposed FedTT framework demonstrates strong performance in both long-term and short-term traffic prediction (i.e., Table 3), underscoring its general advantages in federated traffic transfer knowledge.

## B.5 MODEL SCALABILITY

To evaluate the data volume of target city on the model's performance, we show the traffic flow and speed prediction performance of different methods across different sizes of training data in the target city, ranging from 5% to 40% in the (P8, FT, HK) → P4 scenario using MAE, as shown in Fig. 9. As observed, the FedTT framework consistently achieves the best performance in different-scale datasets with **7.22% to 49.26%** MAE less than

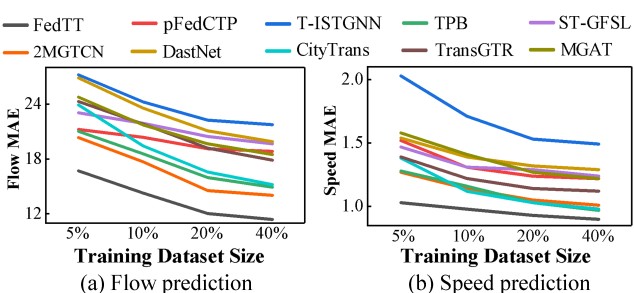

(a) Flow prediction      (b) Speed prediction

Figure 9: Scalability on the data volume of target city

other methods, indicating its superior scalability in FTT. Besides, as the size of the training data increases, all methods exhibit improved performance. This is because more training data enhances the model learning capability on the target city's traffic pattern.

Table 5: Scalability on the number of source cities

| Method | 4 | 8 | 12 | 16 |
|---|---|---|---|---|
| 2MGTCN | 52.19 | 50.31 | 49.03 | 48.65 |
| pFedCTP | 58.74 | 56.27 | 55.15 | 54.82 |
| T-ISTGNN | 67.36 | 66.12 | 65.44 | 65.01 |
| FedTT | 43.25 | 40.88 | 39.52 | 39.97 |

To assess the impact of the number of source cities on the model's performance, we report traffic flow prediction performance using MAE by varying the number of source cities (clients) to 4, 8, 12,

Table 6: Statistics of evaluated cities in UTD19

| City | # instances | # sensors | Interval | Missing Rate |
|------|-------------|-----------|----------|--------------|
| London | 6,454 | 5,719 | 5 min | 19.47% |
| Hamburg | 50,142 | 418 | 3 min | 2.66% |
| Manchester | 6,984 | 181 | 5 min | 10.61% |
| Madrid | 4,560 | 1,116 | 5 min | 16.02% |
| Luzern | 175,116 | 158 | 3 min | 9% |
| Cagliari | 24,000 | 122 | 3 min | 0.59% |
| Marseille | 14,400 | 169 | 3 min | 12.37% |
| Darmstadt | 17,873 | 392 | 3 min | 2.04% |
| Strasbourg | 9,349 | 142 | 3 min | 28.92% |
| Wolfsburg | 6,720 | 133 | 3 min | 0.68% |
| Speyer | 6,714 | 184 | 3 min | 0.2% |
| Bremen | 6,720 | 548 | 3 min | 5.24% |
| Toronto | 5,856 | 188 | 15 min | 14.77% |
| Taipeh | 6,620 | 445 | 3 min | 1.91% |
| Torino | 6,048 | 399 | 5 min | 15.97% |
| Augsburg | 5,757 | 713 | 5 min | 17.64% |
| Groningen | 525 | 55 | 5 min | 1.75% |

and 16, respectively. The statistics of these cities using the UTD19 dataset are provided in Table 6. To ensure temporal consistency across datasets, all traffic data from different cities is uniformly resampled to a 15-minute interval prior to training. For each configuration, we select the first N cities from the list (ordered as in the above table) as source cities and transfer their knowledge to the target city, Groningen. As shown in Table 5, FedTT consistently achieves robust performance and effective knowledge transfer across different numbers of source cities with MAE reduction by **17.13% to 39.61%**, highlighting its scalability and stability in heterogeneous multi-source federated settings. Although all methods exhibit slight performance improvements as the number of source cities increases, the marginal gains gradually diminish, indicating a saturation effect.

## B.6 MODEL EFFICIENCY

Fig. 10 shows the communication size (GB) and running time (minutes). As observed, FedTT has the least communication size and running time compared to other methods, i.e., with communication overhead reduced by **90%** and running time reduced by **1 to 2 orders of magnitude**, showing its superior efficiency. This is because FedTT securely transmits and aggregates the traffic domain-transformed data using the TST module with relatively small

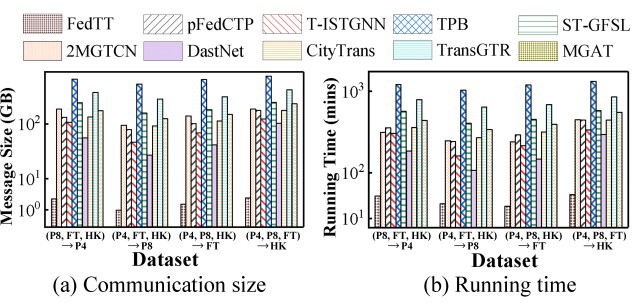

(a) Communication size      (b) Running time

Figure 10: Training efficiency study

computation and communication overheads, compared to other methods that employ homomorphic encryption for model secure aggregation in FTT. Besides, FedTT utilizes the Federated Parallel Training (see Appendix A.1) to train models in parallel, improving the training efficiency.

## B.7 PARAMETER SENSITIVITY

Fig. 11 shows the performance of the FedTT framework with different hyperparameter settings (i.e., $\lambda_1$ and $\lambda_2$) on traffic flow prediction with MAE. First, the suggestion and optimum value of $\lambda_1$ is 0.7. As $\lambda_1$ increases, the generator model tends to generate the data that can "trick" the server discriminator model rather than generating the high-quality traffic domain transformed data, resulting in higher MAE. As $\lambda_1$ decreases, the server dis-

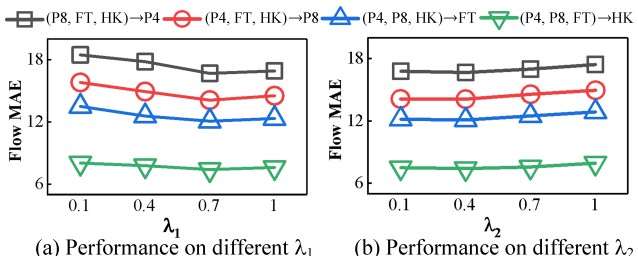

(a) Performance on different $\lambda_1$      (b) Performance on different $\lambda_2$

Figure 11: Parameter sensitivity of FedTT

criminator model loses its ability to effectively guide the generator model in generating traffic domain transformed data, resulting in higher MAE. Second, the suggestion and optimum value of $\lambda_2$ is 0.4. As $\lambda_2$ increases, the generator model tends to generate the data with a traffic domain that deviates significantly from that of the target city, resulting in higher MAE. As $\lambda_2$ decreases, the generator model generates the data with a more local-specific traffic pattern, which hinders the model from effectively learning the traffic patterns of the target city, resulting in higher MAE. Overall, FedTT has the best performance in all hyperparameter settings when $\lambda_1 = 0.7$ and $\lambda_2 = 0.4$, which are used in FedTT as the default values in other experiments.

## B.8 CASE STUDY

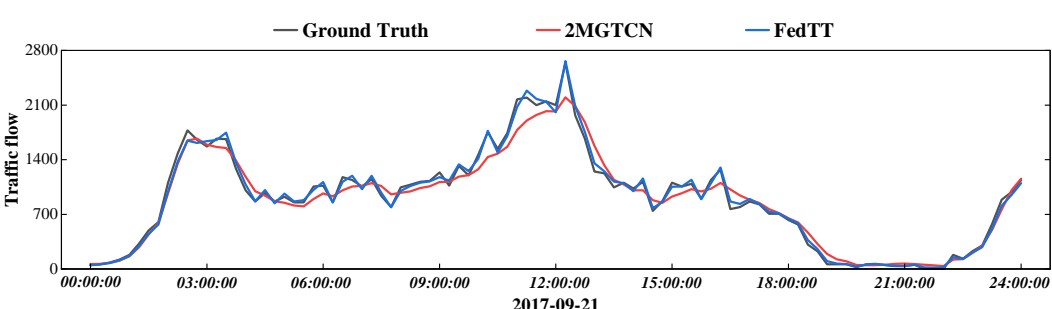

(a) Sensor PGR01_101725_G172_Emmaviaduct_Z_ID_8650_1

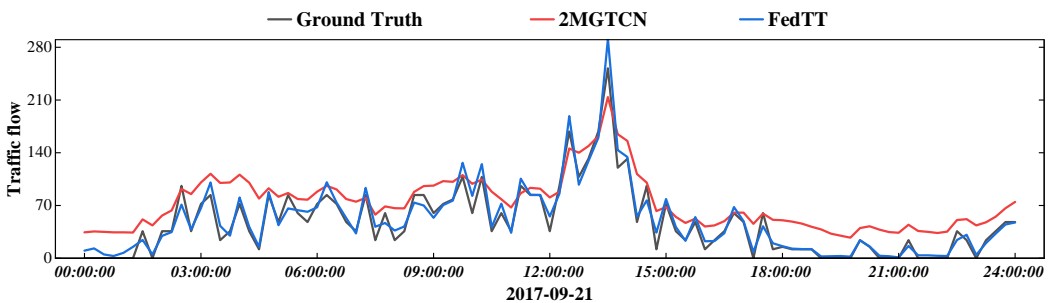

(b) Sensor PGR01_101727_Hereweg_Z_ID_8610_2

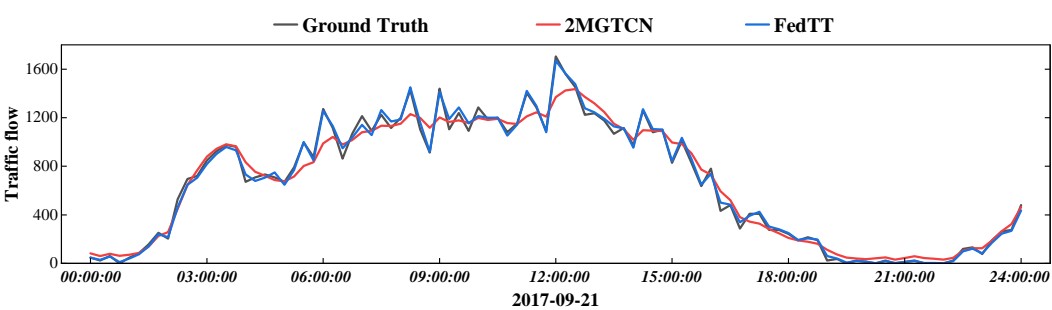

(c) Sensor PGR01_101761_Sontweg_NO_ID_8812_1

Figure 12: Visualization of traffic flow prediction in Groningen

To demonstrate the practical applicability of FedTT in real-world traffic knowledge transfer scenarios, we conduct a case study using the UTD19(Loder et al., 2019) dataset, which includes traffic data from 40 cities worldwide. For comparison, we select 2MGTCN, as it performs the best among the three existing methods in FTT (see Table 3). In this scenario, Groningen is chosen as the target city due to its limited traffic data and relatively sparse sensor deployment, making it challenging to train a high-performance traffic model independently. In contrast, London, Hamburg, Madrid, and Manchester are chosen as source cities because they possess significantly larger datasets and denser

sensor networks, providing abundant traffic data for effective knowledge transfer. The statistics of these cities is summarized in Table 6. Since the sampling intervals of traffic data vary across cities, we resample all datasets in a uniform interval of 15 minutes to ensure that the temporal discrepancies between cities do not affect the model performance.

The traffic flow results of three sensors (i.e., PGR01_101725_G172_Emmaviaduct_Z_ID_8650_1, PGR01_101727_Hereweg_Z_ID_8610_2, and PGR01_101761_Sontweg_NO_ID_8812_1) on September 21, 2017 in Groningen are shown in Fig. 12. As observed, the prediction of FedTT aligns well with the ground truth, while 2MGTCN can only learn the general trend of traffic flow. Taking sensor PGR01_101761_Sontweg_NO_ID_8812_1 as an example. FedTT and 2MGTCN excels from 0:00 a.m. to 6:00 a.m., a period characterized by relatively smooth traffic flow. Throughout the peak hours, from 6 a.m. to 6 p.m., when traffic flow fluctuations are pronounced, FedTT showcases adaptability by learning from the rapid increase and decrease in traffic, while 2MGTCN predicts a relatively smooth traffic flow that does not match the real one. Between 6 p.m. and 12 a.m., as the traffic flow gradually decreases and stabilizes, FedTT maintains relatively accurate predictions compared to 2MGTCN. In summary, the FedTT framework demonstrates its robust performance on real-world traffic knowledge transfer scenarios, yielding satisfactory and accurate prediction results in forecasting the traffic flow across different periods.

## C  LIMITATIONS

Our work has several limitations that merit further investigation. First, the current framework does not incorporate grid-based traffic scenarios, which represents a promising avenue for future research. Second, although our approach is tailored for traffic prediction, its extension to broader spatio-temporal forecasting tasks remains an open challenge. Additionally, the traffic model trained under FedTT for a particular target city cannot be directly transferred to unseen cities, which is a trade-off between model specialization and generalizability. This stems from the inherent duality of urban traffic patterns: global patterns (e.g., daily or weekly periodicities) that are shared across cities, and local patterns (e.g., road topology and traffic regulations) that are city-specific. FedTT is explicitly designed to capture both types of patterns to maximize predictive performance. It not only learns global patterns from source cities but also adapts to the local characteristics of the target city via Traffic Domain Adaptation (TDA). While this design leads to state-of-the-art accuracy in the target city, it limits cross-city generalization, unlike conventional federated traffic training methods that prioritize global patterns at the expense of target-specific accuracy. Finally, while TDA effectively reduces cross-city distribution discrepancies and improves transfer performance, its internal alignment process remains largely opaque. In this work, we focus on designing a practical and privacy-preserving alignment mechanism and evaluating its effectiveness empirically. However, we do not provide a detailed interpretability analysis of how domain representations evolve during alignment or how semantic traffic structures are preserved across cities. Exploring interpretable domain adaptation techniques or visualizing the learned cross-city representations is an important direction for future research and can further enhance the transparency and reliability of FTT in real-world deployments.

## D  LLM USAGE STATEMENT

In the preparation of this manuscript, Large Language Models (LLMs) were used solely as a general-purpose writing assistance tool to improve the clarity, grammar, and fluency of the text. The core research ideas, experimental design, data analysis, and all technical content were conceived and developed entirely by the human authors. No LLM was involved in generating novel ideas, interpreting results, or producing scientific claims. The authors take full responsibility for the accuracy and integrity of all content presented in this paper.

