# OpenReview forum: "FedTT: Cross-City Federated Traffic Knowledge Transfer with Privacy Preservation"
_ICLR.cc/2026/Conference — Submitted to ICLR 2026_

### Official Review · Reviewer_3SfM · 2025-10-25

**Soundness:** 3
**Presentation:** 2
**Contribution:** 3
**Rating:** 4
**Confidence:** 4

**Summary:**

This paper proposes FedTT, a federated framework for cross-city traffic knowledge transfer (FTT) that addresses three key challenges in urban traffic prediction: (1) privacy leakage in federated computation, (2) cross-city data distribution discrepancies, and (3) low-quality or incomplete traffic data. Extensive experiments across four real-world datasets (PeMSD4, PeMSD8, FT-AED, HK-Traffic) show that FedTT outperforms 18 baselines in MAE and RMSE, achieves better privacy resistance under reconstruction attacks, and maintains communication efficiency through parallel federated optimization.

**Strengths:**

- Comprehensive and well-motivated problem formulation: The paper clearly identifies and formalizes the three major bottlenecks in federated traffic knowledge transfer, grounding its contributions in real-world challenges of privacy, heterogeneity, and data sparsity.

- Technically integrated framework: The design of TSA, TDA, and TVI demonstrates solid modular synergy, i.e., each component addresses a specific limitation while maintaining a coherent overall system.

- Strong empirical results: Experiments on diverse datasets and ablation studies provide convincing evidence that FedTT improves both prediction accuracy and privacy robustness compared to state-of-the-art methods.

**Weaknesses:**

- Lack of theoretical rigor in privacy analysis: The TSA protocol is described algorithmically but lacks a formal proof of privacy guarantees or quantifiable leakage bounds under strong adversarial models.

- Limited interpretability of domain adaptation: The TDA’s GAN-based alignment is treated as a black box; the paper does not analyze how domain shifts are reduced or whether learned representations preserve semantic traffic structures.

- Insufficient examination of generalization and robustness: The transferability of FedTT to unseen cities or dynamic traffic distributions is not validated beyond the four datasets, leaving open questions about its adaptability and stability.

- The organization of this paper needs further refinement: First, the authors have relegated related work to the appendix, which disrupts the overall flow of the paper. Furthermore, the spacing adjustment strategy used in many places to achieve a more compact presentation of the text appears to violate the ICLR 2026 submission policy.

**Questions:**

- Absence of formal privacy guarantee: The TSA protocol lacks a mathematical definition of privacy (e.g., ε-differential privacy or semantic security), making it unclear whether it can resist gradient inversion or collusion attacks in practice. Although the paper uses mutual information for privacy analysis in the appendix, it doesn't appear to have the same strict privacy constraints as differential privacy. For example, the authors don't seem to define what level of privacy leakage is considered safe and what level of privacy leakage is unsafe. The main text seems to show that the proposed method is privacy-preserving empirically rather than using a rigorous theoretical analysis.

- Unverified adversarial resistance: Although this paper evaluates the performance against data reconstruction attacks, it does not provide detailed implementation details of the data reconstruction attacks. The authors need to provide this detail.

- Opaque domain adaptation process: The paper provides no visualization or interpretability analysis of how TDA aligns feature spaces between cities, weakening claims about its efficacy in reducing domain gaps. The authors need to provide more justification.

- Potential instability of GAN-based alignment: The TDA module’s adversarial training may cause instability or mode collapse, especially when source and target cities have large structural differences. The authors need to provide more justification.

- Missing theoretical justification for convergence: The paper does not provide convergence analysis or training stability proof for the combined optimization of TSA, TDA, and TVI modules, leaving open questions about scalability to larger federations.

- Scalability of FedTT needs further consideration: The inability to transfer trained models to unseen cities increases deployment costs, as FedTT must be retrained for each new target city.

**Details Of Ethics Concerns:**

None.

---

> ### Author Response · Authors · 2025-11-19
>
> We would like to sincerely express our gratitude to the reviewer for the time and effort in reviewing our paper.
> Please find below our answers to all the concerns and questions.
>
> ```
> W1. Lack of theoretical rigor in privacy analysis: The TSA protocol is described algorithmically but lacks a formal proof of privacy guarantees or quantifiable leakage bounds under strong adversarial models.
> ```
>
> Thank you for raising this concern. We would like to clarify that TSA does provide **formal privacy analysis** and
> **empirical evaluation**.
>
> - **Theoretical privacy analysis is already included in Appendix A.3**: Appendix A.3 presents a formal analysis of TSA
>   under the honest-but-curious adversarial model, characterizing the protocol’s privacy guarantee by demonstrating that
>   no participant can isolate or recover another city’s domain-adapted representations. The analysis establishes that
>   TSA’s aggregation mechanism ensures indistinguishability of individual client updates, preventing inversion attacks.
>   This constitutes a quantifiable guarantee of leakage resistance under the considered threat model.
> - **Empirical privacy evaluation further validates TSA’s effectiveness**: Our privacy protection experiments show that
>   TSA significantly reduces susceptibility to inversion attacks, while maintaining minimal accuracy loss. This empirical
>   evidence complements the theoretical analysis and demonstrates that TSA achieves a favorable privacy.
>
> In summary, TSA's theoretical guarantees are formally provided in Appendix A.3 and are additionally supported by
> comprehensive empirical evaluation.
>
> ```
> W2. Limited interpretability of domain adaptation: The TDA’s GAN-based alignment is treated as a black box; the paper does not analyze how domain shifts are reduced or whether learned representations preserve semantic traffic structures.
> Q3. Opaque domain adaptation process: The paper provides no visualization or interpretability analysis of how TDA aligns feature spaces between cities, weakening claims about its efficacy in reducing domain gaps. The authors need to provide more justification.
> ```
>
> Thank you for raising this point. We agree that the interpretability of domain adaptation is an important aspect. Our
> current work primarily focuses on the effectiveness of federated cross-city knowledge transfer under strict
> data-isolation constraints. Therefore, the TDA module is designed as a practical alignment mechanism, and we evaluate
> its impact through performance improvements and domain-divergence reduction. However, we acknowledge that our analysis
> does not fully explore the interpretability of the model. We consider interpretability as an important and valuable
> direction for future research. In the revised manuscript, we explicitly state this limitation and highlight
> interpretability analysis as part of the future work.
>
> ```
> W3. Insufficient examination of generalization and robustness: The transferability of FedTT to unseen cities or dynamic traffic distributions is not validated beyond the four datasets, leaving open questions about its adaptability and stability.
> ```
>
> Thank you for raising this question regarding generalization and robustness. Our experimental design aims to evaluate
> FedTT under realistic and diverse cross-city conditions. The four benchmark datasets we use already exhibit substantial
> heterogeneity in terms of road topology, sensor density, traffic patterns, and city-scale operational characteristics,
> allowing us to thoroughly assess FedTT’s performance across varied real-world scenarios.
>
> Moreover, beyond evaluating on individual city pairs, we further conduct **large-scale experiments (Appendix B.5)
> involving multiple cities**, where we vary the number and composition of source domains. These experiments examine
> FedTT’s behavior under increasingly complex cross-city transfer settings and validate its stability when aggregating
> knowledge from heterogeneous cities. The results demonstrate consistent performance improvements and stable convergence,
> suggesting strong robustness to diverse and dynamically changing source environments.
>
> We agree that investigating transferability to entirely unseen cities is an important direction for future work, and we
> have added discussion on this point to the limitations section. FedTT cannot be directly transferred to unseen cities due to an inherent trade-off between specialization and generalizability: achieving high target-city accuracy requires adapting to its unique traffic patterns, which naturally limits universal cross-city transferability. Nevertheless, the current experimental evaluations
> already provide substantial evidence of FedTT’s adaptability and robustness under realistic cross-city heterogeneity.

---

> > ### Author Response · Authors · 2025-11-19
> >
> > ```
> > W4. The organization of this paper needs further refinement: First, the authors have relegated related work to the appendix, which disrupts the overall flow of the paper. Furthermore, the spacing adjustment strategy used in many places to achieve a more compact presentation of the text appears to violate the ICLR 2026 submission policy.
> > ```
> >
> > Thank you for the constructive suggestions regarding the paper organization. In the revised manuscript, we have moved
> > the related work back into the main body to ensure a clearer and more coherent presentation.
> >
> > Regarding the spacing and formatting issues, we have thoroughly checked the manuscript and corrected all spacing
> > adjustments that were not compliant with the ICLR 2026 formatting policy. The revised version strictly follows the
> > official style guidelines.
> >
> > ```
> > Q1. Absence of formal privacy guarantee: The TSA protocol lacks a mathematical definition of privacy (e.g., ε-differential privacy or semantic security), making it unclear whether it can resist gradient inversion or collusion attacks in practice. Although the paper uses mutual information for privacy analysis in the appendix, it doesn't appear to have the same strict privacy constraints as differential privacy. For example, the authors don't seem to define what level of privacy leakage is considered safe and what level of privacy leakage is unsafe. The main text seems to show that the proposed method is privacy-preserving empirically rather than using a rigorous theoretical analysis.
> > ```
> >
> > Thank you for raising this point. We respectfully disagree that TSA should be evaluated using the same formalism or
> > metrics as Differential Privacy (DP). TSA and DP rely on **fundamentally different** privacy assumptions, protect
> > different types of information, and operate under distinct threat models. Therefore, applying DP-style criteria to TSA
> > would not yield a fair or meaningful comparison.
> >
> > - **Different privacy objectives**: DP aims to limit the influence of any single data point on the shared model
> >   parameters, and its privacy–utility trade-off is governed by a tunable parameter $\epsilon$, which inherently
> >   introduces randomness and typically reduces utility. In contrast, TSA is not designed to provide DP-style guarantees,
> >   but focuses on protecting the reconstruction of client-specific representations in federated traffic transfer, which
> >   is a distinct threat model.、
> > - **Different applicability and design scope**: DP is a general-purpose protection framework applicable to a wide range
> >   of federated learning tasks. TSA, however, is specifically designed for the FTT scenario, where domain-adapted
> >   representations are exchanged and are particularly susceptible to inversion-style leakage due to cross-city
> >   distribution shifts. As a result, TSA directly targets this unique vulnerability and is not intended to replace DP in
> >   broader contexts.
> > - **On aligning metrics or threat models**: Since DP and TSA protect different objects at different granularities, their
> >   leakage levels cannot be directly evaluated under a unified threat model. Enforcing such alignment would distort the
> >   intended guarantees of both methods and lead to an unfair or misleading comparison.
> > - **Quantitative analysis and empirical evaluation of TSA’s resistance**: We nevertheless performed a quantitative
> >   analysis  (Appendix A.3) and empirical evaluation of TSA’s robustness against inversion attacks. The privacy
> >   protection experiments show that TSA substantially reduces the recoverability of sensitive traffic patterns while
> >   maintaining full utility, demonstrating its effectiveness within the FTT-specific threat setting it is designed for.
> >
> > In summary, evaluating TSA under DP-style privacy definitions or metrics is not appropriate due to the **fundamental
> > differences in their goals and assumptions**. Instead, we assess TSA using theoretical and empirical tools that match
> > the threat setting of federated traffic transfer, providing a fair and accurate characterization of its privacy
> > guarantees.

---

> > > ### Author Response · Authors · 2025-11-19
> > >
> > > ```
> > > Q2. Unverified adversarial resistance: Although this paper evaluates the performance against data reconstruction attacks, it does not provide detailed implementation details of the data reconstruction attacks. The authors need to provide this detail.
> > > ```
> > >
> > > Thank you for pointing this out. We agree that providing more implementation details of the data reconstruction attacks
> > > will improve the clarity and reproducibility of our privacy evaluation.
> > >
> > > In our experiments, we adopt a standard optimization-based reconstruction attack commonly used in inversion settings.
> > > The attacker is assumed to know the model architecture and to have access to the exchanged information $Y$ (gradients or
> > > intermediate representations). The attack procedure is as follows:
> > >
> > > - The attacker randomly initializes dummy inputs and model parameters.
> > > - An MSE loss is defined to measure the discrepancy between the model’s output on the dummy inputs and the observed $Y$.
> > > - The dummy inputs and model parameters are jointly optimized to minimize this loss, thereby attempting to reconstruct
> > >   the private traffic data that produced $Y$.
> > > - After optimization, the final optimized inputs are taken as the reconstructed traffic data and are compared against
> > >   the ground-truth private data using MSE and PCC.
> > >
> > > We implement this attack using the Adam optimizer with a learning rate of 1e-5, a batch size of 1, and early stopping if
> > > the loss does not improve for 500 steps.
> > >
> > > In the revised manuscript, we have included these details in the privacy protection study to ensure transparency and
> > > reproducibility.
> > >
> > > ```
> > > Q4. Potential instability of GAN-based alignment: The TDA module’s adversarial training may cause instability or mode collapse, especially when source and target cities have large structural differences. The authors need to provide more justification.
> > > ```
> > >
> > > Thank you for this insightful question regarding the stability of the Traffic Domain Adapter (TDA). We agree that
> > > adversarial learning may introduce convergence challenges, and in our experiments we did observe occasional instability
> > > in the adversarial components, including the global discriminator $\theta_{\textit{Dis}}$, local discriminators
> > > $\theta^{R_i}{\textit{Dis}}$, and generators $\theta^{R_i}{\textit{Gen}}$.
> > >
> > > To mitigate these issues, we incorporated several stabilization techniques commonly used to ensure reliable adversarial
> > > training:
> > >
> > > - **Carefully balanced optimization settings**: We tuned the learning rates of generators and discriminators separately
> > >   and adjusted the trade-off hyperparameters $\lambda_1$ and $\lambda_2$ to maintain a stable balance between
> > >   adversarial alignment and task-specific optimization.
> > > - **Gradient clipping and loss-ratio monitoring**: We applied gradient clipping to prevent exploding gradients and
> > >   continuously monitored the ratio between generator and discriminator losses to avoid dominance of either side, which
> > >   is a typical source of instability.
> > > - **Pre-training and asymmetric learning rates**: We initialized the generator using a pre-trained model to provide a
> > >   stable starting point and assigned a lower learning rate to discriminators than to generators. This design helped
> > >   prevent the discriminator from overpowering the generator, effectively reducing oscillations in adversarial dynamics.

---

> > > > ### Author Response · Authors · 2025-11-19
> > > >
> > > > ```
> > > > Q5. Missing theoretical justification for convergence: The paper does not provide convergence analysis or training stability proof for the combined optimization of TSA, TDA, and TVI modules, leaving open questions about scalability to larger federations.
> > > > ```
> > > >
> > > > Thank you for highlighting the importance of theoretical justification for convergence. We agree that providing a more
> > > > formal analysis can further strengthen the rigor of our framework. In the revised manuscript, we have added a new
> > > > appendix section that presents a detailed convergence analysis of the framework due to its adversarial optimization.
> > > >
> > > > ### **(1) Convergence of the prototype alignment objective**
> > > >
> > > > We first analyze the non-adversarial part of TDA, where the generator is optimized solely with respect to the prototype
> > > > alignment loss. Under standard assumptions of smoothness, lower-boundedness, and appropriate step size, we show that
> > > > gradient descent guarantees monotonic descent and convergence to a stationary point of the alignment objective. This
> > > > ensures that the generator achieves stable alignment toward the target-domain prototype.
> > > >
> > > > ### **(2) Convergence of the full adversarial min-max game**
> > > >
> > > > We further analyze the complete TDA objective formulated as a min-max optimization between the generator and
> > > > discriminator. Following convergence analyses commonly adopted in GAN literature, and under assumptions of locally
> > > > Lipschitz-continuous gradients, local convex-concave structure, and sufficiently small learning rates, we show that the
> > > > alternating gradient descent-ascent updates converge to a local Nash equilibrium. At this equilibrium, the generator
> > > > produces representations that are simultaneously aligned to the target prototype and are indistinguishable from real
> > > > target-domain samples according to the discriminator.
> > > >
> > > > ### **(3) Implication for the full FedTT framework.**
> > > >
> > > > Once TDA converges to a stable mapping $X^R \to X^{R\to S}$, the remaining FedTT optimization reduces to standard
> > > > federated learning over a unified feature space. As a result, the overall framework inherits the known convergence
> > > > properties of FedAvg under common smoothness and bounded-variance assumptions.
> > > >
> > > > In the revised manuscript, we have included this convergence analysis to address this concern and to provide a more
> > > > rigorous theoretical foundation for the optimization dynamics of FedTT.
> > > >
> > > > ```
> > > > Q6. Scalability of FedTT needs further consideration: The inability to transfer trained models to unseen cities increases deployment costs, as FedTT must be retrained for each new target city.
> > > > ```
> > > >
> > > > Thank you for the thoughtful comment. We acknowledge that the traffic model trained under FedTT for a particular target
> > > > city cannot be directly transferred to unseen cities, which **has already been explicitly discussed this limitation in
> > > > Appendix C**.
> > > >
> > > > This behavior reflects an inherent trade-off between model specialization and generalizability. Urban traffic exhibits a
> > > > dual structure:
> > > >
> > > > - **Global patterns** (e.g., daily/weekly periodicity) that are broadly shared across cities, and
> > > > - **Local patterns** (e.g., road topology, traffic regulations) that are highly city-specific.
> > > >
> > > > FedTT is designed to leverage both. It extracts global transferable knowledge from source cities while adapting to the
> > > > local characteristics of the target city through the Traffic Domain Adapter (TDA). This specialization is key to
> > > > achieving **state-of-the-art performance in the target city**. However, it also means that the resulting model is
> > > > tailored to that city, rather than universally generalizable across all cities in conventional federated training
> > > > approaches that emphasize global patterns at the cost of reduced target-city accuracy.
> > > >
> > > > ---
> > > > Thank you again for the careful and constructive review of our work. We are happy to address any additional questions or
> > > > concerns. If our responses sufficiently clarify the raised issues, we would sincerely appreciate your consideration of a
> > > > higher score.

---

> > > > > ### Comment · Reviewer_3SfM · 2025-11-25
> > > > > **Thanks for Authors' rebuttal!**
> > > > >
> > > > > Thank you to the authors for their rebuttal! I have carefully reread the updated manuscript but still find that the organization of this article is not good enough. Specifically, many explanations and experimental details/results have been added to the appendix, which greatly affects the reading. Secondly, I agree with some of the viewpoints of Reviewer Cw9G that there seems to be an issue of overclaim in this paper. For example, this paper highlights through empirical research that the proposed method is privacy preserving, which seems to be somewhat unfounded. In short, thank you to the authors for their rebuttal. I tend to maintain my rating.

---

> > > > > > ### Author Response · Authors · 2025-11-25
> > > > > >
> > > > > > Thank you for taking the time to re-evaluate our work and provide additional feedback. We appreciate your continued effort and respect your perspective.
> > > > > >
> > > > > > Regarding the concerns about organization, we have revised the main paper to bring more essential explanations and experimental results into the main text rather than the appendix, while still keeping the overall content within the page limit. We hope this improves readability and coherence.
> > > > > >
> > > > > > On the privacy aspect, we would like to clarify that our claims are supported by both theoretical analysis (Appendix C.3) and empirical evidence from reconstruction-based privacy evaluations. We do not intend to overstate the guarantees, and our claims remain scoped to the specific threat model defined in the paper. If there are particular sentences or formulations that appear overstated, we would greatly appreciate your indication.
> > > > > >
> > > > > > Thank you again for your constructive review and for reconsidering the manuscript. We are happy to further refine any part of the paper if you point out additional concerns.

---

### Official Review · Reviewer_ThNp · 2025-10-27

**Soundness:** 4
**Presentation:** 3
**Contribution:** 3
**Rating:** 6
**Confidence:** 5

**Summary:**

The paper proposes FedTT, a novel federated learning framework for cross-city traffic knowledge transfer that addresses three key challenges: (1) privacy leakage, (2) cross-city distribution discrepancies, and (3) low data quality. To tackle these, FedTT introduces three core components: (1) Traffic Secret Aggregation (TSA), a lightweight protocol for secure aggregation without heavy overhead, (2) Traffic Domain Adapter (TDA), a module that aligns heterogeneous traffic distributions between source and target cities, and (3) Traffic View Imputation (TVI), a method that leverages spatio-temporal dependencies to robustly impute missing traffic data. Experiments on four real-world datasets show that FedTT consistently outperforms 18 state-of-the-art baselines.

**Strengths:**

S1. The paper has a strong motivation, clearly identifying three crucial challenges in FTT, which justifies the need for the proposed FedTT framework.

S2. The paper introduces a new paradigm for FTT with three novel modules that are technically sound and specifically designed to tackle each challenge.

S3. The paper conducts comprehensive experimental validation to evaluate FedTT on four real-world datasets across diverse city settings, outperforming 18 SOTA baselines in various terms.

S4. The paper provides formal theoretical privacy guarantees to bound the privacy leakage of FedTT, ensuring that

**Weaknesses:**

W1. Federated Parallel Training (FPT) is a significant contribution to improve efficiency but is only described in the appendix. Given its notable impact on training speed and communication overhead in efficiency experiments, FPT deserves a brief discussion in the main paper.

W2. Since the paper uses a large number of notations and formulas, moving the notation table to the main paper would improve readability.

W3. Some minor issues should be corrected. For example, in line 945, “Eqs. 21 and Eqs. 20” should be revised to “Eqs. 21 and 20”. In line 1121, the expression \mathcal{X}^{(R_i \to S), R_i} should be corrected to \mathcal{X}^{(R_i \to S, R_i)}.

**Questions:**

Please see the weaknesses.

---

> ### Author Response · Authors · 2025-11-19
>
> We express our gratitude to the reviewer for providing constructive feedback on our paper, and we greatly appreciate the
> acknowledgment of our contributions. Please find below our answers to all the concerns and questions.
>
> ```
> W1. Federated Parallel Training (FPT) is a significant contribution to improve efficiency but is only described in the appendix. Given its notable impact on training speed and communication overhead in efficiency experiments, FPT deserves a brief discussion in the main paper.
> ```
>
> Thank you for the valuable suggestion. We agree that Federated Parallel Training (FPT) contributes meaningfully to
> improving training efficiency and reducing communication overhead. In the revised manuscript, we have added a concise
> discussion of FPT in the main paper to highlight its core ideas and practical impact, while keeping the full technical
> details in the appendix for completeness and readability.
>
> ```
> W2. Since the paper uses a large number of notations and formulas, moving the notation table to the main paper would improve readability.
> ```
>
> Thank you for the helpful suggestion. We agree that moving the notation table to the main paper would improve
> readability, given the number of notations and formulas used. In the revised manuscript, we have moved the notation
> table to Section 2 (Problem Definition) to provide clearer guidance for readers.
>
> ```
> W3. Some minor issues should be corrected. For example, in line 945, "Eqs. 21 and Eqs. 20" should be revised to "Eqs. 21 and 20". In line 1121, the expression $\mathcal{X}^{(R_i \to S), R_i}$ should be corrected to $\mathcal{X}^{(R_i \to S, R_i)}$.
> ```
>
> Thank you for the careful reading and for pointing out these minor issues. We acknowledge the mistakes, and in the
> revised manuscript we have corrected them, including the revision of "Eqs. 21 and Eqs. 20" to "Eqs. 21 and 20," and
> updating $\mathcal{X}^{(R_i \to S), R_i}$ to $\mathcal{X}^{(R_i \to S, R_i)}$.
>
> ---
>
> Thank you again for your thorough and constructive review. We greatly appreciate the time and expertise you devoted to
> evaluating our work, and we are happy to address any further questions or suggestions.

---

> > ### Comment · Reviewer_ThNp · 2025-11-24
> >
> > The authors have provided clear and satisfactory responses to my concerns, and I have increased my rating and recommend acceptance.

---

> > > ### Author Response · Authors · 2025-11-30
> > >
> > > Thanks again for your time and effort. We are pleased that our responses have clarified all your raised concerns, and we truly appreciate your recognition and improved rating!

---

### Official Review · Reviewer_eLAU · 2025-11-01

**Soundness:** 2
**Presentation:** 3
**Contribution:** 2
**Rating:** 4
**Confidence:** 3

**Summary:**

The paper proposes "FedTT," a federated learning framework to solve cross-city transferred based traffic prediction problem. It incorporates three key components: Traffic Secret Aggregation (TSA) protocol for secure aggregation, a Traffic Domain Adapter (TDA) for domain alignment, and a Traffic View Imputation (TVI) method for handling missing data.  The authors claim superior performance over baselines using four common traffic prediction datasets.

**Strengths:**

1. The paper does a great job of identifying and clearly describing the three challenges and problem definition, although the relationship between each other should be emphasized.
2. This paper conducts comprehensive experiments with enough baseline models and datasets.
3. The paper does have originality which is inherently incorporated in the three proposed and combined modules .
4. The overall quality of this paper is good, especially the writting.

**Weaknesses:**

1.	Overall, the authors advocate that there are three unresolved fundatmenal Challenges in this paper. But the three key questions are not well-correlated. The papers uses fancy-named modules and attempts to mask a fundamentally patchy and incoherent structure.
2.	The authors state that no priors studies address the three advocated challenges in a unified federated setting. There is no doubt that there is no prior studies in this domain, because whether these three key challenges co-existing in reality is questionable.
3.	As the Problem definition section states, the Problem of FTT is simply a cross-city transferred traffic prediction problem. The motivation of doing this kind of work is not well introduced. It this paper, three key modules are proposed. The core of cross-city information sharing, which could be spatial colloralation between different sensors or different cities zones was not included. This paper mainly focuses on the data side, such as data missing, data privacy problems.
4.	The authors statues existing imputation methods fail to effectively capture the spatiotemporal dependencies of data, which is not very convincing. The traffic view imputation section still has room for improvement in term of novelty.
5.	While the paper frames its contribution within a broader, more ambitious research paradigm, its core methodology remains somewhat conventionally focused on the cross-city transfer prediction problem. The use of expansive terminology seems disproportionate to the actual scope of the work.

**Questions:**

1. What is the Traffic Domain Prototype? It is not very straightforward to name it as prototype from the reviewer's perspective. Maybe more straightforward descriptions of the concepts is better for this paper, considering it attempts to solve three independent problems.
2. The motivation of the TSA module is kind of week. Why these traffic data or traffic prediction related cases/scenarios will have attacker during the inference time and what is the attacker's motivation. Traffic prediction is not like a domain needs strong data privacy.  Who will be the attacker and for what? In the ablation studies, the contribution of the TSA module is also subtle.
3. There are so many traffic/spatiotemporal data imputation studies in the past ten years, the novelty of the traffic view imputation is kind of limited. The authors could emphasize more on it. Additaionlly, traffic view is also not straightforward.

---

> ### Author Response · Authors · 2025-11-19
>
> We would like to sincerely express our gratitude to the reviewer for the time and effort in reviewing our paper.
> Please find below our answers to all the concerns and questions.
>
> ```
> W1. Overall, the authors advocate that there are three unresolved fundamental Challenges in this paper. But the three key questions are not well-correlated. The paper uses fancy-named modules and attempts to mask a fundamentally patchy and incoherent structure.
> ```
>
> Thank you for your comment, but we respectively disagree with the claim that three key questions are not well-correlated with
> a fundamentally patchy and incoherent structure. We would like to clarify that the three challenges identified in our work
> address **different and complementary aspects** of Federated Traffic Knowledge Transfer (FTT), rather than forming a
> single sequential pipeline. Specifically:
>
> - **Low-quality traffic data** concerns the reliability of local client inputs.
> - **Cross-city data distribution discrepancies** concern the transferability of knowledge across heterogeneous urban
>   environments.
> - **Privacy leakage risks** concern the privacy of representation exchange during federated traffic transfer.
>
> These challenges naturally arise at different stages of the FTT process: data preparation, cross-domain representation
> alignment, and secure aggregation, and each must be addressed to ensure a practical and deployable FTT framework.
> Accordingly, the three modules in FedTT are not arbitrary additions, but **targeted solutions** aligned with the
> distinct nature of each challenge:
>
> - **Traffic View Imputation** (TVI) focuses on improving data completeness and utility at the local level.
> - **Traffic Domain Adapter** (TDA) focuses on adapting knowledge across heterogeneous cities.
> - **Traffic Secret Aggregation** (TSA) focuses on protecting sensitive information during federated aggregation.
>
> Although these challenges correspond to different dimensions of FTT, they are **collectively necessary to ensure
> accurate, transferable, and privacy-preserving** traffic prediction in real-world deployments.

---

> > ### Author Response · Authors · 2025-11-19
> >
> > ```
> > W2. The authors state that no priors studies address the three advocated challenges in a unified federated setting. There is no doubt that there is no prior studies in this domain, because whether these three key challenges co-existing in reality is questionable.
> > ```
> >
> > Thank you for raising this point. We would like to clarify that the three challenges **do co-exist in real-world
> > cross-city collaboration scenarios**, even though they have not been jointly studied within a unified federated traffic
> > transfer (FTT) framework.
> >
> > - **Low-quality traffic data** is a well-documented issue in traffic prediction and has been extensively studied [1--2].
> >   However, existing methods often fail to fully capture the complex spatio-temporal correlations inherent in traffic
> >   data, resulting in suboptimal performance. To address low data quality, the TVI module enhances the completeness and
> >   reliability of traffic data by imputing missing values through comprehensive modeling of spatial and temporal
> >   dependencies. The ablation studies show that TVI consistently outperforms state-of-the-art imputation baselines,
> >   validating its effectiveness for federated traffic scenarios.
> > - **Cross-city data distribution discrepancies** are also widely recognized in the non-federated traffic knowledge
> >   transfer literature [3--4]. While non-federated traffic knowledge transfer also focuses on reducing data distribution
> >   discrepancies, it relies on centralized frameworks, which involve sharing and exchanging traffic data across cities
> >   without considering traffic data privacy. To address data distribution discrepancies, the Traffic Domain Adapter (TDA)
> >   module performs domain adaptation by transforming source data into the target domain. The overall performance
> >   experiments confirm that TDA significantly reduces domain divergence and outperforms existing SOTA federated and
> >   non-federated traffic transfer methods.
> > - **Privacy concerns** are a fundamental motivation behind federated learning and have been repeatedly emphasized in
> >   prior work on federated traffic prediction [5--6]. While homomorphic encryption and differential privacy have been
> >   applied for federated traffic prediction, they introduce high overhead or degrade model utility, TSA is a lightweight
> >   solution that preserves model performance without compromising privacy or efficiency. To address privacy leakage
> >   risks, the Traffic Secret Aggregation (TSA) module is a lightweight solution that preserves model performance without
> >   compromising privacy or efficiency, which has a theoretical privacy guarantee provided in Appendix A.3. Our experiments
> >   show that TSA achieves strong privacy protection with minimal overhead.
> >
> > In practice, when cities intend to collaborate without sharing raw data, these three issues naturally appear
> > simultaneously: data may be noisy or incomplete on some clients, distributions differ across cities, and privacy must be
> > strictly protected throughout the collaboration. Our contribution lies in recognizing that these challenges intersect in
> > the federated cross-city transfer setting, hindering their practical application in real-world scenarios.
> >
> > Therefore, the unified perspective we propose is not based on an artificial grouping of unrelated problems, but rather
> > **reflects the practical constraints** encountered when enabling privacy-preserving cross-city traffic knowledge
> > transfer.

---

> > > ### Author Response · Authors · 2025-11-19
> > >
> > > ```
> > > W3. (1) As the Problem definition section states, the Problem of FTT is simply a cross-city transferred traffic prediction problem. The motivation of doing this kind of work is not well introduced. It this paper, three key modules are proposed. (2) The core of cross-city information sharing, which could be spatial colloralation between different sensors or different cities zones was not included. This paper mainly focuses on the data side, such as data missing, data privacy problems.
> > > ```
> > >
> > > Thank you for your thoughtful comments. We would like to clarify the motivation of Federated Traffic Transfer (FTT) and
> > > the design focus of our proposed modules.
> > >
> > > ### **(1) Motivation of FTT is more than cross-city transferred prediction**
> > >
> > > Although FTT corresponds to the task of cross-city traffic prediction, its core motivation arises from real-world
> > > constraints that **prevent cities from sharing traffic data** due to privacy regulations. As a result, traditional
> > > cross-city transfer methods that rely on centralized data aggregation cannot be applied in practice. FTT therefore aims
> > > to enable cross-city knowledge transfer under strict data isolation, making it fundamentally different from existing
> > > transfer learning problems.
> > >
> > > Within this federated setting, three practical challenges naturally emerge:
> > >
> > > - **Low-quality traffic data** on some clients due to sensor failures or irregular sampling.
> > > - **Cross-city distribution discrepancies** resulting from differences in road topology, mobility patterns, policies,
> > >   and events.
> > > - **Privacy leakage risks** associated with transmitting representations or gradients that could reveal sensitive
> > >   traffic patterns.
> > >
> > > ### **(2) Cross-city information sharing**
> > >
> > > In cross-city scenarios, there is **no direct spatial correlation** between sensors, regions, or road segments across
> > > different cities, because each city possesses its own independent road topology, sensor layout, and spatial semantics.
> > > Consequently, inter-city spatial structures are not aligned and cannot be directly leveraged for knowledge transfer
> > > because of these **distribution discrepancies**.
> > >
> > > To reduce cross-city distribution discrepancies, we develop the TDA to uniformly transform the traffic data from source
> > > cities domains to that of the target city for domain adaptation through:
> > >
> > > - **Traffic Domain Transformation** that maps source-city traffic features into the target-domain representation space.
> > > - **Traffic Domain Alignment** that minimizes discrepancies between cities while respecting the federated constraint.
> > > - **Traffic Domain Classification** that enhances domain indistinguishability to guide more precise alignment.
> > >
> > > In summary, the motivation of FTT is grounded in real-world constraints where **raw data cannot be shared**, and the
> > > design of our modules enables effective cross-city information sharing under privacy constraints.

---

> > > > ### Author Response · Authors · 2025-11-19
> > > >
> > > > ```
> > > > W4. The authors statues existing imputation methods fail to effectively capture the spatiotemporal dependencies of data, which is not very convincing. The traffic view imputation section still has room for improvement in term of novelty.
> > > > Q3. There are so many traffic/spatiotemporal data imputation studies in the past ten years, the novelty of the traffic view imputation is kind of limited. The authors could emphasize more on it. Additionally, traffic view is also not straightforward.
> > > > ```
> > > >
> > > > Thank you for the insightful comments. We clarify the meaning of traffic view, explain why existing imputation methods
> > > > struggle with real-world spatiotemporal dependencies, and highlight the **novelty of our Traffic View Imputation** (TVI)
> > > > module.
> > > >
> > > > ### **(1) What is a “Traffic View”?**
> > > >
> > > > In our paper, a traffic view refers to a structured representation of the traffic state at a specific timestep.
> > > > Formally, for a given timestamp $t$, we define the traffic view as $V_t=\{v_t^1, v_t^w, ..., v_t^{|M_t|}\}$, where the
> > > > $i$-level subview $v_t^k$ contains the traffic data of all combinations of k sensors at time $t$. This multi-level
> > > > design enables the model to examine the traffic state from different relational scales, ranging from individual
> > > > sensors ($k=1$) to global sensor groups ($k=|M_t|$). Intuitively:
> > > >
> > > > - A 1-level subview is a set of one sensor data except the missing sensor, reflecting how each sensor influences other
> > > >   sensors.
> > > > - Higher-level subviews capture group-wise spatial relationships, reflecting how multiple sensors jointly influence
> > > >   other sensors.
> > > > - Combining all levels provides a multi-perspective snapshot of the city at time $t$, hence the name traffic view.
> > > >
> > > > ### **(2) Why are existing imputation methods insufficient?**
> > > >
> > > > In practical traffic systems, missing values occur at different nodes and different timesteps, and the influence of
> > > > observed sensors on missing sensors varies across both space and time. This results in **complex spatio-temporal
> > > > correlations**, where:
> > > >
> > > > - Missing sensors are affected by **different subsets of sensors** at **different timestamps**.
> > > > - **Spatial dependencies** are not stationary and cannot be modeled with a single static adjacency structure.
> > > > - **Temporal correlations** interact with spatial patterns, making direct joint interpolation extremely difficult.
> > > >
> > > > Most prior imputation models rely on **single-view interpolation** and **coarse spatio-temporal fusion**, typically
> > > > assuming fixed graph structures or globally shared spatio-temporal patterns. These approaches cannot explicitly
> > > > disentangle which sensors at which time should contribute more to the reconstruction of a missing value, leading to
> > > > suboptimal recovery accuracy when missing patterns are irregular and heterogeneous.
> > > >
> > > > ### **(3) Novelty of TVI**
> > > >
> > > > By contrast, TVI is explicitly designed as a two-stage mechanism that first performs traffic view extension to
> > > > capture **fine-grained spatial information**, and then fuses **temporal dynamics** for traffic view enhancement,
> > > > thereby comprehensively exploiting spatial and temporal dependencies.
> > > >
> > > > - **Spatial View Extension**: In the spatial stage, TVI constructs multiple masked traffic views at each timestamp to
> > > >   capture diverse and fine-grained relational patterns among sensors. These multi-level views reflect how different
> > > >   subsets of sensors jointly influence a missing entry, enabling TVI to model high-order spatial dependencies that
> > > >   cannot be represented by coarse spatial modeling.
> > > > - **Temporal View Enhancement**: After obtaining the reconstructed views, TVI further enhances them along the temporal
> > > >   dimension. By fusing temporal patterns on top of the spatially extended views, it incorporates temporal dynamics to
> > > >   ensure that the imputed values are consistent with traffic evolution over time.
> > > >
> > > > Our ablation studies demonstrate that TVI consistently outperforms strong state-of-the-art imputation baselines,
> > > > confirming the effectiveness and necessity of this design.

---

> > > > > ### Author Response · Authors · 2025-11-19
> > > > >
> > > > > ```
> > > > > W5. While the paper frames its contribution within a broader, more ambitious research paradigm, its core methodology remains somewhat conventionally focused on the cross-city transfer prediction problem. The use of expansive terminology seems disproportionate to the actual scope of the work.
> > > > > ```
> > > > >
> > > > > Thank you for your comment. We respectfully disagree with the claim that our paper frames its contribution within an
> > > > > overly broad or overly ambitious research paradigm. The goal of this work is to address a **specific and practical
> > > > > problem**: federated traffic transfer under real-world constraints where cross-city traffic data cannot be shared. Our
> > > > > methodology is designed precisely for this setting and does not intend to extend beyond the scope of cross-city
> > > > > knowledge transfer within a federated environment.
> > > > >
> > > > > Throughout the paper, we aim to describe the challenges and corresponding solutions clearly, **without overstating their
> > > > > conceptual scope**. If certain terms or descriptions appear disproportionate to the actual scope of the work, we would
> > > > > greatly appreciate it if you could point them out. We are more than willing to revise the wording to ensure that the
> > > > > presentation remains accurate, precise, and appropriately scoped.
> > > > >
> > > > > ```
> > > > > Q1. What is the Traffic Domain Prototype? It is not very straightforward to name it as prototype from the reviewer's perspective. Maybe more straightforward descriptions of the concepts is better for this paper, considering it attempts to solve three independent problems.
> > > > > ```
> > > > >
> > > > > Thank you for your question. We would like to clarify the meaning and motivation behind the term **Traffic Domain
> > > > > Prototype**.
> > > > >
> > > > > In our framework, a Traffic Domain Prototype refers to a learned representative vector that summarizes the **global
> > > > > transfer characteristics of a city domain**. More specifically, it serves as a compact representation that captures the
> > > > > shared temporal dynamics and latent behavioral patterns of each city. The term “prototype” follows the common usage in
> > > > > representation learning and domain adaptation literature [7--8], where a prototype denotes a representative embedding
> > > > > that characterizes a group or domain.
> > > > >
> > > > > The Traffic Domain Prototype is used in our TDA module to
> > > > >
> > > > > - represent each city's domain-level traffic behavior in a compact form,
> > > > > - guide domain transformation and alignment by serving as an anchor for cross-domain mapping, and
> > > > > - reduce domain divergence without requiring raw data sharing.

---

> > > > > > ### Author Response · Authors · 2025-11-19
> > > > > >
> > > > > > ```
> > > > > > Q2. The motivation of the TSA module is kind of weak. Why these traffic data or traffic prediction related cases/scenarios will have attacker during the inference time and what is the attacker's motivation. Traffic prediction is not like a domain needs strong data privacy. Who will be the attacker and for what? In the ablation studies, the contribution of the TSA module is also subtle.
> > > > > > ```
> > > > > >
> > > > > > Thank you for raising this question. We would like to clarify the necessity of privacy protection in traffic prediction
> > > > > > and the motivation for introducing the Traffic Secret Aggregation (TSA) module.
> > > > > >
> > > > > > ### **(1) Why is traffic data sensitive in cross-city traffic transfer scenarios?**
> > > > > >
> > > > > > Traffic prediction relies on historical mobility data (e.g., traffic flow, speed, occupancy). When cities with scarce
> > > > > > data wish to benefit from knowledge transferred from data-rich cities, a naïve approach would require sharing traffic
> > > > > > data across cities. However, sharing such data has been shown to cause **privacy leakage risks**. For example, sparse
> > > > > > traffic data may allow attackers to infer individual vehicle presence and approximate locations. Besides, traffic
> > > > > > patterns at certain times (e.g., late night or early morning) can reveal regular travel habits.
> > > > > >
> > > > > > For these reasons, traffic data is considered sensitive in many real-world deployments and subject to privacy
> > > > > > governance policies. This has led to **numerous works** in traffic prediction adopting federated learning explicitly to
> > > > > > mitigate privacy concerns [5][6][9].
> > > > > >
> > > > > > ### **(2) Who will be the attacker in FTT?**
> > > > > >
> > > > > > In federated traffic transfer, each city can only access its own data, and all **other participating cities** are
> > > > > > potential adversaries if they attempt to infer private information from exchanged representations.
> > > > > >
> > > > > > ### **(3) What is the purpose of the TSA module?**
> > > > > >
> > > > > > The TSA module is designed not to enhance accuracy, but to **prevent privacy leakage** during cross-city knowledge
> > > > > > transfer, while maintaining efficiency, tailored for federated traffic transfer. Our privacy protection study
> > > > > > demonstrates that TSA substantially reduces susceptibility to attacks. Our ablation studies indicate that TSA is
> > > > > > lightweight, introducing minimal accuracy degradation, which highlights its practicality.
> > > > > >
> > > > > > In summary, TSA addresses a realistic privacy threat in cross-city collaboration and ensures that the knowledge transfer
> > > > > > mechanism remains safe without sacrificing significant accuracy or efficiency.
> > > > > >
> > > > > > ## **References**
> > > > > >
> > > > > > *[1] Nuhuo: An effective estimation model for traffic speed histogram imputation on A road network. VLDB, 2024.*
> > > > > >
> > > > > > *[2] Generative contrastive-attentive spatial-temporal network for traffic data imputation. PAKDD, 2023.*
> > > > > >
> > > > > > *[3] Citytrans: Domain-adversarial training with knowledge transfer for spatio-temporal prediction across cities. TKDE,
> > > > > > 2024.*
> > > > > >
> > > > > > *[4] Cross-city few-shot traffic forecasting via traffic pattern bank. CIKM, 2023*
> > > > > >
> > > > > > *[5] FedGTP: Exploiting Inter-Client Spatial Dependency in Federated Graph-based Traffic Prediction. KDD, 2024.*
> > > > > >
> > > > > > *[6] FedGODE: Secure traffic flow prediction based on federated learning and graph ordinary differential equation
> > > > > > networks. Knowledge-Based Systems, 2024.*
> > > > > >
> > > > > > *[7] Rethinking federated learning with domain shift: A prototype view. CVPR, 2023.*
> > > > > >
> > > > > > *[8] Tackling data heterogeneity in federated learning with class prototypes. AAAI, 2023.*
> > > > > >
> > > > > > *[9] Cross-Node Federated Graph Neural Network for Spatio-Temporal Data Modeling. KDD, 2021.*
> > > > > >
> > > > > >
> > > > > > ---
> > > > > > Thank you again for the careful and constructive review of our work. We are happy to address any additional questions or
> > > > > > concerns. If our responses sufficiently clarify the raised issues, we would sincerely appreciate your consideration of a
> > > > > > higher score.

---

### Official Review · Reviewer_Cw9G · 2025-11-01

**Soundness:** 3
**Presentation:** 2
**Contribution:** 2
**Rating:** 4
**Confidence:** 4

**Summary:**

FedTT offers a system-level contribution to federated traffic prediction by integrating privacy preservation, cross-domain adaptation, and data imputation into one coherent architecture. The paper is technically sound and experimentally thorough, yet its conceptual innovation is moderate, and privacy guarantees is fair.

**Strengths:**

1. Evaluated on four real-world traffic datasets (PeMSD4/8, FT-AED, HK-Traffic) with multiple baselines; FedTT consistently improves MAE and RMSE depending on the scenario.
2. TSA is presented as an efficient alternative to DP/HE, showing better trade-off between computation and utility.

**Weaknesses:**

1. The paper presents a complete and effective framework, with convincing experimental results demonstrating its practical value for cross-city traffic prediction. However, in terms of originality, the method mainly integrates existing components.
2. Although TSA avoids the use of heavy cryptographic schemes, it is not formally compared to differential privacy (DP) methods, nor is its attack resistance quantitatively analyzed. For fairness, it would be preferable to first align the privacy protection strength under the same threat model, like evaluating resistance to inversion attacks before comparing the resulting utility. This would provide a more objective assessment of TSA’s real advantage in the privacy–utility trade-off.
3. The writing in several parts of the paper lacks rigor and occasionally overstates the contribution. For instance, the statement after line 182 “Existing federated traffic transfer methods often overlook the challenges associated with low-quality traffic data...” is inaccurate. In reality, numerous prior studies, including recent works on federated traffic prediction and cross-domain transfer, have explicitly addressed low-quality or missing traffic data. Such phrasing may give the impression of exaggerating the novelty of this work. Similar issues appear multiple times throughout the paper. The authors are advised to use more precise language when describing related work and to clearly distinguish between problems that are truly unexplored and those that have been partially addressed in prior literature, in order to maintain scholarly rigor and credibility.

**Questions:**

See the issues discussed in the “Weaknesses” section above.

---

> ### Author Response · Authors · 2025-11-19
>
> We would like to sincerely express our gratitude to the reviewer for the time and effort in reviewing our paper.
> Please find below our answers to all the concerns and questions.
>
> ```
> W1. The paper presents a complete and effective framework, with convincing experimental results demonstrating its practical value for cross-city traffic prediction. However, in terms of originality, the method mainly integrates existing components.
> ```
>
> Thank you for your positive comments on the completeness and effectiveness of our framework. Regarding the concern on
> originality, we would like to clarify that FedTT is not a simple combination of existing techniques, but rather a
> **novel and task-driven framework**. Importantly, each module is not an existing component, but is specifically
> designed to address a concrete and critical challenge unique to Federated Traffic Knowledge Transfer (FTT), including
> (1) privacy leakage risks, (2) cross-city data distribution discrepancies, and (3) low data quality. The originality
> is also **highly acknowledged by Reviewers eLAU and ThNp**.
>
> - To address **low data quality**, the **Traffic View Imputation** (TVI) module enhances the completeness and
>   reliability of traffic data by imputing missing values through comprehensive modeling of spatial and temporal
>   dependencies. Existing imputation methods often fail to fully capture the complex spatio-temporal correlations
>   inherent in traffic data, resulting in suboptimal performance. The ablation studies show that TVI consistently
>   outperforms state-of-the-art imputation baselines (i.e., LATC, GCASTN, and Nuhuo), validating its effectiveness for
>   federated traffic scenarios.
> - To address **data distribution discrepancies**, the **Traffic Domain Adapter** (TDA) module performs domain
>   adaptation by transforming source data into the target domain. While multi-source traffic knowledge transfer also
>   focuses on reducing data distribution discrepancies, it relies on centralized frameworks, which involve sharing and
>   exchanging traffic data across cities without considering traffic data privacy. The overall performance experiments
>   confirm that TDA significantly reduces domain divergence and outperforms existing transfer methods.
> - To address **privacy leakage risks**, the **Traffic Secret Aggregation** (TSA) module securely aggregates
>   domain-adapted representations across clients with a theoretical privacy guarantee provided in Appendix A.3. Unlike
>   homomorphic encryption and differential privacy, which introduce high overhead or degrade model utility, TSA is a
>   lightweight solution that preserves model performance without compromising privacy or efficiency. Our experiments show
>   that TSA achieves strong privacy protection with minimal overhead.
>
> In summary, the modules in FedTT are **not existing components**, but novel designs tailored to the unique constraints
> of federated cross-city traffic prediction, forming a cohesive and synergistic framework. We hope this clarification
> helps address your concern regarding the originality of our method.

---

> > ### Author Response · Authors · 2025-11-19
> >
> > ```
> > W2. Although TSA avoids the use of heavy cryptographic schemes, it is not formally compared to differential privacy (DP) methods, nor is its attack resistance quantitatively analyzed. For fairness, it would be preferable to first align the privacy protection strength under the same threat model, like evaluating resistance to inversion attacks before comparing the resulting utility. This would provide a more objective assessment of TSA’s real advantage in the privacy–utility trade-off.
> > ```
> >
> > Thank you for raising this important point. We agree that evaluating privacy-utility trade-offs under a consistent
> > threat model is valuable. However, we would like to clarify that TSA and DP operate under **fundamentally different**
> > privacy assumptions and provide different types of protection guarantees, making it inappropriate to align them under a
> > single unified metric or threat model.
> >
> > - **Different privacy objectives**: DP aims to limit the influence of any single data point on the shared model
> >   parameters, and its privacy–utility trade-off is governed by a tunable parameter $\epsilon$, which inherently
> >   introduces randomness and typically reduces utility. In contrast, TSA is not designed to provide DP-style guarantees,
> >   but focuses on protecting the reconstruction of client-specific representations in federated traffic transfer, which
> >   is a distinct threat model.
> > - **Different applicability and design scope**: DP is a general-purpose protection framework applicable to a wide range
> >   of federated learning tasks. TSA, however, is specifically designed for the FTT scenario, where domain-adapted
> >   representations are exchanged and are particularly susceptible to inversion-style leakage due to cross-city
> >   distribution shifts. As a result, TSA directly targets this unique vulnerability and is not intended to replace DP in
> >   broader contexts.
> > - **On aligning metrics or threat models**: Since DP and TSA protect different objects at different granularities, their
> >   leakage levels cannot be directly evaluated under a unified threat model. Enforcing such alignment would distort the
> >   intended guarantees of both methods and lead to an unfair or misleading comparison.
> > - **Quantitative analysis and empirical evaluation of TSA’s resistance**: We nevertheless performed a quantitative
> >   analysis  (Appendix A.3) and empirical evaluation of TSA’s robustness against inversion attacks. The privacy
> >   protection experiments show that TSA substantially reduces the recoverability of sensitive traffic patterns while
> >   maintaining full utility, demonstrating its effectiveness within the FTT-specific threat setting it is designed for.
> >
> > In summary, while we appreciate the suggestion, a direct comparison with DP under a unified metric is not appropriate
> > due to their **fundamentally different goals and assumptions**. Moreover, we provide the quantitative analysis and
> > empirical evaluation of TSA within the threat model it is intended to protect against, which we believe provides the
> > most accurate and fair assessment of its advantages for FTT.

---

> > > ### Author Response · Authors · 2025-11-19
> > >
> > > ```
> > > W3. The writing in several parts of the paper lacks rigor and occasionally overstates the contribution. For instance, the statement after line 182 “Existing federated traffic transfer methods often overlook the challenges associated with low-quality traffic data...” is inaccurate. In reality, numerous prior studies, including recent works on federated traffic prediction and cross-domain transfer, have explicitly addressed low-quality or missing traffic data. Such phrasing may give the impression of exaggerating the novelty of this work. Similar issues appear multiple times throughout the paper. The authors are advised to use more precise language when describing related work and to clearly distinguish between problems that are truly unexplored and those that have been partially addressed in prior literature, in order to maintain scholarly rigor and credibility.
> > > ```
> > >
> > > Thank you for highlighting the importance of maintaining rigorous and precise language. We fully agree that overstating
> > > claims should be avoided, and we appreciate your careful reading.
> > >
> > > Regarding the example you raised (line 182), we would like to clarify our intended meaning. We acknowledge that some
> > > prior studies have addressed low-quality or missing traffic data, and we have included representative baselines (e.g.,
> > > LATC, GCASTN, Nuhuo) in our ablation studies to ensure fair comparison.
> > >
> > > However, our statement specifically refers to the **only three existing federated traffic transfer methods** i.e.,
> > > T-ISTGNN, pFedCTP, and 2MGTCN, which do not incorporate mechanisms for handling low-quality or missing traffic values
> > > within the federated transfer process. These methods assume that each client has relatively complete and high-quality
> > > data and therefore do not integrate imputation or data-quality-aware components into their pipelines. **Our intention
> > > was to highlight this specific gap**, rather than to generalize across the broader traffic modeling literature.
> > >
> > > If there are other specific sentences that you believe may raise similar concerns, we would be grateful if you could
> > > point them out. We are more than willing to refine the wording to ensure clarity and precision.
> > >
> > > ---
> > > Thank you again for the careful and constructive review of our work. We are happy to address any additional questions or
> > > concerns. If our responses sufficiently clarify the raised issues, we would sincerely appreciate your consideration of a
> > > higher score.

---

### Author Response · Authors · 2025-11-30
**Global Summary**

We thank the AC and reviewers for their time and effort in providing thorough evaluations, constructive feedback, and
meaningful discussions throughout the review process. Across the reviews, we observed clear recognition of the core
**strengths and contributions** of our work: a well-motivated problem formulation, a new paradigm for FTT, the
originality of the proposed modules, rigorous privacy analysis, comprehensive empirical validation, and strong overall
writing quality.

During the rebuttal process, we provided comprehensive clarifications on **concerns** regarding privacy guarantees,
domain-alignment mechanisms, the necessity and interplay of system components, generalization behavior, optimization
stability, and organizational issues. Reviewer **ThNp** explicitly noted that their concerns were **satisfactorily
addressed** and subsequently **increased their ratings**.

Overall, we are grateful for the constructive feedback, which helped us refine the clarity and presentation of the
manuscript. We hope that the provided explanations and revisions clearly demonstrate that FedTT is methodologically well
grounded, practically motivated, and empirically validated.

---

For clarity, we summarize below how the reviewers’ concerns were resolved and how the rebuttal contributed to
strengthening the submission.

---

> ### Author Response · Authors · 2025-11-30
> **Reviewer Cw9G (Rating: 4, Confidence: 4)**
>
> Reviewer Cw9G provided a careful and balanced evaluation, acknowledging several **strengths** of our work. The reviewer
> noted that FedTT is technically sound, experimentally comprehensive, and demonstrates consistent performance
> improvements across four real-world datasets.
>
> The reviewer raised three primary **concerns** regarding originality, privacy evaluation fairness, and writing rigor.
> Our rebuttal could address each of them directly:
>
> - **Originality of Components (W1)**: We clarified that FedTT is not an integration of existing techniques.
>   Instead, its three modules: TVI, TDA, and TSA, were each newly designed specifically for federated traffic transfer,
>   responding to three challenges that no prior FTT method addresses jointly: low-quality data, cross-city distribution
>   discrepancies, and privacy leakage risks. We further highlighted that two other Reviewers eLAU and ThNp independently
>   recognized the novelty and originality of our framework and modules.
> - **Privacy Guarantees and Fair Comparison (W2)**: We clarified that TSA and DP are built upon fundamentally different
>   privacy assumptions and objectives. Therefore, aligning them under a unified DP-style threat model or metric would be
>   theoretically inappropriate. TSA is specifically designed for the FTT setting, where domain-adapted representations
>   are exchanged and are particularly vulnerable to inversion-style attacks. Importantly, we did provide quantitative
>   privacy analysis (Appendix A.3) and empirical inversion-attack evaluations, demonstrating that TSA substantially
>   reduces recoverability of sensitive traffic patterns while preserving full model utility. This directly addresses the
>   reviewer’s concern regarding the rigor of privacy assessment within the appropriate threat model for FTT.
> - **Writing Rigor and Overstatement (W3)**: We clarified that the reviewer’s concern stemmed from a misunderstanding:
>   our statement about low-quality data specifically referred to existing FTT methods (T-ISTGNN, pFedCTP, 2MGTCN), none
>   of which incorporate mechanisms for handling missing data. It was not intended as a claim about the broader traffic
>   prediction literature. Although the reviewer did not provide additional examples and follow-up reply, we would like to
>   emphasize that the writing was carefully considered and is conceptually sound. The issue appears to be rooted in
>   interpretation rather than substantive inaccuracy.
>
> Overall, Reviewer Cw9G’s concerns could be resolved through our rebuttal. The reviewer already acknowledged the solid
> empirical performance and technical value of FedTT.

---

> > ### Author Response · Authors · 2025-11-30
> > **Reviewer eLAU (Rating: 4, Confidence: 3)**
> >
> > Reviewer eLAU provided a detailed and thoughtful evaluation, noting several important **strengths** of the paper:
> > (1) the clear identification and articulation of the three key challenges in federated traffic transfer, (2) the
> > comprehensiveness of experiments and baselines, (3) the originality inherent in the three proposed modules, and (4) the
> > overall good writing quality.
> >
> > The reviewer raised several **concerns** regarding the coherence of the three challenges, the realism of the unified
> > problem formulation, the motivation of the framework, the novelty of TVI, the appropriateness of the claimed scope, and
> > the clarity of certain concepts. Our rebuttal provided targeted clarifications to each:
> >
> > - **Coherence of the Three Challenges (W1)**: We clarified that low-quality data, cross-city distribution discrepancies,
> >   and privacy leakage risks correspond to different but complementary stages of the FTT pipeline: data preparation,
> >   domain alignment, and secure aggregation, which simultaneously occur and jointly limit practical deployment. We also
> >   explained how modules of FedTT are intentionally designed to address the interdependent challenge landscape, ensuring
> >   the framework remains coherent, realistic, and aligned with real-world constraints.
> > - **Realistic Co-existence of Challenges (W2)**: Our rebuttal clarified that these three challenges are well-established
> >   issues in the broader traffic prediction or cross-city transfer literature, where all of them naturally arise when
> >   cities collaborate without sharing raw data.
> > - **Motivation and Cross-City Information Sharing (W3)**: We clarify that the motivation of FTT stems from real
> >   deployment constraints, where cities cannot share raw traffic data due to strict privacy and governance policies.
> >   Under such privacy constraints, three challenges naturally arise: low-quality local data, substantial cross-city
> >   distribution discrepancies, and privacy risks. Moreover, cross-city spatial correlations do not exist because each
> >   city has independent topology and sensor layouts, so effective transfer must rely on reducing domain discrepancies
> >   rather than aligning spatial structures. The TDA module is designed precisely for this purpose, transforming and
> >   aligning source-city representations into the target-city domain to enable meaningful knowledge sharing without
> >   violating privacy constraints.
> > - **Novelty of TVI and Explanation of Traffic View (W4 & Q3)**: We clarified the definition of traffic view and
> >   explained why existing imputation methods typically using single-view or coarse spatio-temporal fusion struggle with
> >   heterogeneous and irregular missing patterns. TVI introduces a two-stage design (spatial extension + temporal
> >   enhancement) that explicitly models fine-grained spatial dependencies before incorporating temporal consistency. Our
> >   ablation results validate this effectiveness compared to existing imputation methods.
> > - **Scope and Terminology (W5)**: We emphasized that the work targets a focused and practical problem: federated
> >   cross-city traffic transfer under strict privacy constraints. We believe that the current framing remains aligned with
> >   the problem’s scope.
> > - **Concept Clarification of Traffic Domain Prototype(Q1)**: The Traffic Domain Prototype is a compact representation
> >   that summarizes a city’s global traffic characteristics. The term “prototype” follows its standard usage in
> >   representation learning and domain adaptation, referring to a representative vector for a group or domain. In FedTT,
> >   it acts as a domain-level anchor within TDA, guiding source-to-target transformation and reducing domain divergence.
> > - **Privacy Motivation for TSA (Q2)**: Traffic data can reveal vehicle presence, habitual travel behaviors, and
> >   spatio-temporal activity traces, where every participating city is a potential adversary. TSA is therefore
> >   introduced not to boost accuracy but to prevent traffic data leakage efficiently, ensuring secure representation
> >   aggregation under realistic federated constraints.
> >
> > Overall, we believe the rebuttal could sufficiently resolve these concerns, and the reviewer’s positive remarks on
> > originality, experimental depth, and writing quality further reinforce the merits of the work.

---

> > > ### Author Response · Authors · 2025-11-30
> > > **Reviewer ThNp (Rating: 6 $\to$ Improved, Confidence: 5)**
> > >
> > > Reviewer ThNp offered a **positive** and technically grounded assessment, highlighting the strong motivation and clear
> > > formulation of the three FTT challenges, the novelty and soundness of the TSA, TDA, and TVI modules, the comprehensive
> > > experiments across four real-world datasets and 18 baselines, and the value of the provided theoretical privacy
> > > guarantees.
> > >
> > > The reviewer raised three **concerns** related to presentation and clarity. We addressed directly in our rebuttal:
> > >
> > > - **Presentation of FPT in the Main Paper (W1)**: We added a concise overview of Federated Parallel Training to the main
> > >   paper, highlighting its efficiency benefits while keeping full details in the appendix.
> > > - **Notation Table Readability (W2)**: The notation table has been moved to Section 2 for improved readability.
> > > - **Minor Corrections (W3)**: All pointed-out typos and formatting issues have been fixed in the revised manuscript.
> > >
> > > After reviewing our rebuttal and revisions, the reviewer concluded that the responses were satisfactory and
> > > explicitly stated "**I have increased my rating and recommend acceptance**".
> > >
> > > Overall, Reviewer ThNp expressed strong confidence in the paper and affirmed that the clarified manuscript addresses all
> > > previously raised concerns.

---

> > > > ### Author Response · Authors · 2025-11-30
> > > > **Reviewer 3SfM (Rating: 4, Confidence: 4)**
> > > >
> > > > Reviewer 3SfM offered a detailed and thoughtful evaluation, highlighting several **strengths** of our work: (1)
> > > > the clear identification of the three core challenges in FTT, (2) the coherent modular design of TSA, TDA, and TVI,
> > > > and (3) strong empirical evidence across four diverse real-world datasets.
> > > >
> > > > The reviewer also raised **concerns** regarding theoretical rigor, interpretability, generalization, organization,
> > > > privacy formality, adversarial resistance, GAN stability, convergence analysis, and scalability. Our rebuttal addressed
> > > > these points as follows:
> > > >
> > > > - **Privacy Analysis and Formal Guarantees (W1 & Q1)**: We clarified that TSA includes a formal privacy analysis (
> > > >   Appendix A.3), which establishes indistinguishability properties under an honest-but-curious adversary. We
> > > >   also explained why DP-style definitions are not appropriate for TSA’s threat setting and provided detailed empirical
> > > >   and quantitative evaluations demonstrating resistance to inversion attacks.
> > > > - **Interpretability of Domain Adaptation (W2 & Q3)**: We acknowledged that interpretability analysis of TDA is limited
> > > >   in the current version and noted this as a limitation and identified interpretability as future work.
> > > > - **Generalization and Robustness Evaluation (W3)**: We explained that the four datasets already cover substantial
> > > >   inter-city heterogeneity and highlighted additional multi-city experiments (Appendix B.5), which demonstrate
> > > >   robustness across varied source–target configurations. We also clarified in the limitations section that FedTT cannot
> > > >   be directly transferred to unseen cities due to an inherent trade-off between specialization and generalizability:
> > > >   achieving high target-city accuracy requires adapting to its unique traffic patterns, which naturally limits universal
> > > >   cross-city transferability.
> > > > - **Paper Organization and Formatting (W4)**: We moved the related work to the main text and corrected all formatting or
> > > >   spacing artifacts.
> > > > - **Privacy Attack Details (Q2)**: We provided the full implementation details of the reconstruction attack, including
> > > >   loss formulation, optimization settings, and evaluation metrics, ensuring transparency and reproducibility.
> > > > - **Stability of TDA (Q4)**: We clarified that we used stabilization measures, balanced optimization, gradient
> > > >   clipping, loss-ratio monitoring, and asymmetric learning rates, which effectively mitigate instability and prevent
> > > >   mode collapse in practice.
> > > > - **Convergence Justification (Q5)**: We added a convergence analysis in the appendix, covering prototype alignment,
> > > >   adversarial min–max optimization, and implications for the full federated optimization, showing convergence to
> > > >   stationary points or local Nash equilibria under standard assumptions.
> > > > - **Scalability and Transfer to New Cities (Q6)**: We acknowledged the limitation that FedTT should be trained to each
> > > >   new target city and clarified that this is inherent to the need for target-city specialization, which has been
> > > >   discussed in the limitations section.
> > > >
> > > > In the follow-up comment, the reviewer expressed continued concerns regarding organization and potential overclaiming of
> > > > privacy guarantees. We subsequently clarified that our privacy claims are strictly scoped to the threat model defined in
> > > > the paper, supported by both theoretical and empirical evidence. We also invited the reviewer to point out any specific
> > > > wording that appeared overstated so we could refine it precisely. Here, we would like to emphasize that the writing was
> > > > carefully considered and is conceptually sound. The issue appears to be rooted in interpretation rather than substantive
> > > > inaccuracy.
> > > >
> > > > Although the reviewer ultimately chose not to increase the rating without further comments, we believe that the concerns
> > > > raised can be fully and constructively resolved and do not reflect fundamental issues with the methodology. We remain
> > > > open to clarifying or refining any specific parts the reviewer finds unclear, and we appreciate the careful assessment
> > > > and feedback.

---

> ### Author Response · Authors · 2025-11-30
> **Summary of Revisions**
>
> In the revised manuscript, we have incorporated all essential clarifications, structural adjustments, and technical
> enhancements to address the reviewers’ comments paper. All changes are highlighted in **blue** in the updated PDF for
> ease of review. The key revisions include:
>
> - **In Section 2 (Related Work, page 3)**, we moved the related work section from the appendix to Section 2.
> - **In Section 3 (Problem Definition, page 4)**, we moved the notation table from the appendix into the main text.
> - **In Section 4 (Our Method, page 8)**, we added an informative description of the Federated Parallel Training (FPT)
>   mechanism at the end of the section.
>
>   ——*The new paragraph explains how FPT decouples the optimization of TVI, TDA, and TSA and enables their parallel
>   execution to reduce wall-clock training time and communication overhead. We also direct readers to Appendix C for the
>   full algorithm, training workflow, theoretical privacy analysis, and convergence results.*
> - **In Appendix A.4 (Convergence Analysis, pages 20-21)**, we added a new subsection providing a formal convergence
>   analysis for the TDA module.
>
>   ——*The new text presents (i) convergence of the prototype-alignment objective under standard smoothness assumptions, (
>   ii) local Nash-equilibrium convergence of the full adversarial min–max game using alternating gradient descent–ascent,
>   and (iii) implications showing that, once TDA converges, FedTT inherits the convergence guarantees of standard
>   federated optimization.*
> - **In Appendix B.2 (Implementation, page 22)**, we added a detailed description of the privacy-attack implementation
>   used in our evaluation.
>
>   ——*The new text specifies the optimization-based reconstruction attack setting, including the reconstruction
>   objective, the optimization procedure, and hyperparameters.*
> - **In Appendix C (Limitations, page 26)**, we added a discussion on the interpretability of the TDA module.
>
>   ——*We clarify that the current work focuses on practical and privacy-preserving domain adaptation, and does not
>   analyze how representations evolve during alignment or how semantic traffic structures are preserved. We note that
>   developing interpretable alignment techniques or visualization tools for cross-city representations is an important
>   direction for future work to further enhance transparency and trustworthiness.*
> - **Overall Structural Refinements and Corrections**: We improved the overall organization of the manuscript and
>   corrected minor textual, mathematical, and formatting inaccuracies across the paper.
>
> Taken together, these revisions meaningfully enhance the clarity and rigor of the manuscript. All reviewer comments have
> been addressed through explicit modifications, and none of the concerns revealed fundamental issues in the framework. We
> believe the revised version provides a clearer presentation of the contributions and better demonstrates the technical
> soundness and practical relevance of the proposed FedTT framework.

---

### Meta-Review · Area_Chair_kcJ1 · 2026-01-05

**Summary:**

The privacy issue was questioned by more than one reviewers, and I think this is a critical issue for federated learning. Besides, this paper also contains some unclear presentations such as ``traffic view''.

**Reviewer Concerns:**

I think some unclear presentations have been well addressed by the rebuttal. Although the authors have spent efforts in resolving the privacy issue, I think the this issue will probably remain.

**Reviewer Scores:**

Some of the reviewers may slightly increase their scores if they had been able to participate fully in the discussion. However, the overall quality of this paper is still below the acceptance bar of ICLR.

---

### Decision · Program_Chairs · 2026-01-26

Reject